# FORGETTING IS EVERYWHERE

## ABSTRACT

A fundamental challenge in developing general learning algorithms is their tendency to forget past knowledge when adapting to new data. Addressing this problem requires a principled understanding of forgetting; yet, despite decades of study, no unified definition has emerged that provides insights into the underlying dynamics of learning. We propose an algorithm- and task-agnostic theory that characterises forgetting as a lack of self-consistency in a learner's predictive distribution over future experiences, manifesting as a loss of predictive information. Our theory naturally yields a general measure of an algorithm's propensity to forget. To validate the theory, we design a comprehensive set of experiments that span classification, regression, generative modelling, and reinforcement learning. We empirically demonstrate how forgetting is present across all learning settings and plays a significant role in determining learning efficiency. Together, these results establish a principled understanding of forgetting and lay the foundation for analysing and improving the information retention capabilities of general learning algorithms.

## 1 INTRODUCTION

Forgetting is a ubiquitous yet poorly understood phenomenon in machine learning (McCloskey & Cohen, 1989). When a learner updates its beliefs based on new observations, it often forgets prior knowledge. This leads to a degradation in performance on previous observations. Although this behaviour is well documented in continual learning (CL; Kirkpatrick et al., 2016) and reinforcement learning (RL; Atkinson et al., 2021; Khetarpal et al., 2022), forgetting also occurs in independent and identically distributed (i.i.d.) settings (Lee & Storkey, 2023).

Most studies of forgetting come from the CL literature. Here, metrics usually track how performance degrades on earlier tasks after training on later tasks (Chaudhry et al., 2018a). Although widely adopted, these measures poorly capture forgetting and often conflate two distinct phenomena: *backward transfer*, where new learning improves performance on past tasks (Benavides-Prado & Riddle, 2022), and *forgetting*, where updates erode prior knowledge (Jagielski et al., 2022). This makes it challenging to distinguish between constructive and destructive adaptation.

In contrast, we study forgetting as a *fundamental property of learning*. It is a consequence of how any adaptive system updates its beliefs. CL, RL, and neural networks are not our focus. Instead, we treat them as instances of a broader phenomenon. We believe that before effective learning algorithms for CL can be developed, a precise conceptual understanding of forgetting must first be formulated. Therefore, our aim is to provide a general formalism that characterises forgetting in learning systems.

This motivates a new conceptual foundation of forgetting, built on the following insight:

> *If a learner updates its predictions on data it already expects, that update cannot represent the acquisition of new information. Instead, it must represent the loss of previously acquired knowledge.*

This allows us to give a precise and general definition of forgetting. We ask: *What is forgetting? When and why does it occur? How does forgetting impact learning?*

We address these questions through the following contributions:

1. **Formulation of learning over time**: Inspired by Hutter (2005); Dong et al. (2022); Fortini & Petrone (2019); Fong et al. (2023), we define a *general theoretical formulation* for reasoning about how learners acquire, retain, and lose capabilities during learning (§3).

2. **Conceptualisation of forgetting**: We formulate forgetting from a *predictive perspective*. Specifically, we define forgetting as a violation of *self-consistency* in the learner's predictive distribution, providing a unified probabilistic foundation that generalises prior conceptions of forgetting (§4.2).

3. **Operational measure**: This conceptualisation naturally yields a measure of the propensity to forget, which we propose to assess the validity of our formalism (Definition 4.6).

4. **Empirical characterisation**: We empirically study the propensity to forget in diverse environments and learning paradigms. These include regression, classification, generative modelling, CL, and RL. Our results confirm that forgetting dynamics conform to expected characteristics. They also reveal a trade-off between forgetting and training efficiency (§5).

## 2 RELATED WORK

In this paper, we study the prevalence of forgetting in general learning algorithms. While terminology varies across fields, all phenomena involving the loss of previously acquired knowledge reflect the same underlying process. For clarity, we refer to all such phenomena collectively as *forgetting*.

**Forgetting in CL.** Forgetting is often studied in the context of CL (De Lange et al., 2021; Wang et al., 2024), where various proxy metrics measure forgetting in sequential learning settings, often as the loss of performance on previous tasks (Kirkpatrick et al., 2016; Chaudhry et al., 2018b;a).

However, this approach cannot distinguish between two distinct effects: *backwards transfer*, where learning on new data improves performance on past tasks (Benavides-Prado & Riddle, 2022), and forgetting, where learning on new data degrades previously acquired knowledge (McCloskey & Cohen, 1989; Jagielski et al., 2022). Performance on previous tasks depends on both how current data informs the learner about past data (*backwards transfer*) and the extent to which updating on new tasks results in a loss of previous knowledge. Furthermore, most metrics are limited to CL settings and do not generalise to all learners or domains, motivating the need for a more general definition of forgetting.

Various approaches have sought to quantify forgetting. In toy settings with known data distributions, forgetting can be measured relative to an oracle (Lee et al., 2021). Others examine changes in internal representations (Kim et al., 2025) or model the trade-off between generalisation and forgetting (Raghavan & Balaprakash, 2021). Mechanistic accounts in associative memory models demonstrate how interference between correlated patterns leads to retrieval errors (Hopfield, 1982; Amit et al., 1985). A related line of work studies models trained on their own generated outputs, typically in the context of generative model collapse (Alemohammad et al., 2023; Shumailov et al., 2023; Bertrand et al., 2023; Scholten et al., 2025). These studies implicitly touch on the phenomenon we formalise as forgetting: updating a model on self-generated targets can degrade previously encoded knowledge.

**Forgetting in RL.** Forgetting has also been studied in RL. Early work in continual RL noted the risk of knowledge degradation when agents adapt over long time horizons (Ring, 1994; 1997). Recent surveys highlight that forgetting is a persistent challenge in RL (Khetarpal et al., 2022).

Empirical studies document several forms of forgetting, with forgetting measured in a manner similar to the CL counterpart, either by tracking the degradation of performance on previously learned tasks or by quantifying the distributional shift in the agent's policy or value function over time (Shenfeld et al., 2025). Value-based methods with function approximation often lose performance on earlier estimates (Mnih et al., 2015; van Hasselt et al., 2018). This is closely related to the phenomenon of *policy churn* identified by (Schaul et al., 2022) in which the greedy policy of a value-based learner changes in a large portion of the input space after just a few updates. Policy gradient methods are also prone to overwriting earlier strategies during continual adaptation (Ring, 1994; Kirkpatrick et al., 2017). Replay buffers are widely used to mitigate these effects by reintroducing past experiences during training (Rolnick et al., 2019; Fedus et al., 2020).

**Misconceptions of forgetting.** Throughout the literature, we identify a general trend: forgetting is often characterised through a mechanism-specific lens. Model-centric views equate forgetting with parameter drift (McCloskey & Cohen, 1989; French, 1999; Rusu et al., 2016; Kirkpatrick et al., 2017; Zenke et al., 2017; Aljundi et al., 2018; Masse et al., 2018; Li et al., 2024; Zhao et al., 2023), or policy drift (Shenfeld et al., 2025), while accuracy-centric views characterise forgetting as performance decay on earlier tasks (Kemker et al., 2018; Parisi et al., 2019; Jagielski et al., 2022; Van de Ven et al., 2022).

This fragmentation has hindered the development of coherent principles for understanding and mitigating forgetting. Moreover, these definitions mischaracterise forgetting and are constrained by specific models or tasks. Not all parameter or policy changes imply forgetting: not all learners have parameters, yet in those that do, parameters can change without the learner necessarily forgetting anything. We demonstrate this in §5.1 with a learner whose parameters change without causing forgetting.

Motivated by these limitations, we treat the predictive distribution as the object of interest. We propose a principled conceptualisation of forgetting based on predictive self-consistency that applies broadly across learning paradigms. We further demonstrate the generality of our conceptualisation and show how it supports an empirical measure of forgetting that aligns with theoretical expectations.

## 3 LEARNING AND INFERENCE PROCESSES

We present a general framework in which supervised learning, RL, and generative modelling are all specific cases of a single stochastic interaction process. The formalism draws inspiration from the agent-environment perspective from general RL (Hutter, 2005; Lattimore, 2014; Dong et al., 2022; Abel et al., 2023; Hutter et al., 2024; Kumar et al., 2025), but is adapted to emphasise conventions standard in machine learning (Bishop, 2006; Goodfellow et al., 2016; Fong et al., 2023).

**Notation.** We let capital calligraphic letters denote measurable spaces $(\mathcal{X})$[1], lowercase letters denote elements $(x)$ or functions $(f)$, and uppercase italics denote random variables $(X)$. For any measurable space $\mathcal{X}$, we let $\mathcal{D}(\mathcal{X})$ denote the set of probability distributions over $\mathcal{X}$. For example, a mapping

$$p : \mathcal{X} \times \mathcal{Y} \to \mathcal{D}(\mathcal{Z}) \tag{1}$$

is interpreted as a family of conditional distributions $p(\cdot \mid x, y)$ on $\mathcal{Z}$, indexed by $(x, y) \in \mathcal{X} \times \mathcal{Y}$.

### 3.1 PRELIMINARIES: LEARNING INTERACTION

We formalise learning as an ongoing interaction between a *learner* and an *environment*, evolving over discrete time steps $t \in \mathbb{N}_0 = \{0, 1, \dots\}$. We distinguish between two sources of stochasticity:

- *External probabilities* $p_e(\cdot)$ describe the stochasticity inherent to the environment.
- *Predictive distributions* $q_f(\cdot)$ describe the predictive uncertainty of the learner.

To formalise this interaction, we define a structure that describes the data exchanged between the learner and the environment.

**Definition 3.1** (Interface). An *environment-learner interface* is a pair $(\mathcal{X}, \mathcal{Y})$ of measurable spaces.

Elements of $\mathcal{X}$ are *observations*, encompassing the signals the environment emits (e.g., observations, targets, etc.), and elements of $\mathcal{Y}$ are *outputs*, encompassing the signals emitted by the learner (e.g., actions, classifications, regression values, etc.).

**Definition 3.2** (Histories). The set of *histories* relative to an interface $(\mathcal{X}, \mathcal{Y})$ is

$$\mathcal{H} = \bigcup_{t=0}^{\infty} (\mathcal{X} \times \mathcal{Y})^{t+1}. \tag{2}$$

We refer to an element of $\mathcal{H}$ as a history $H \in \mathcal{H}$, which is a sequence $H_{0:t} = ((X_0, Y_0), \dots, (X_t, Y_t))$ of $t + 1$ observation-output pairs $(X_i, Y_i)$. For a sequence $A = (A_0, A_1, \dots, A_t)$ and indices $0 \le i \le j \le t$, we denote the subsequence between indices $i$ and $j$ as $A_{i:j} = (A_i, A_{i+1}, \dots, A_j)$.

**Definition 3.3** (Environment). An *environment* relative to interface $(\mathcal{X}, \mathcal{Y})$ is a pair $(e, p_{X_0})$, where

- $e : \mathcal{H}, \mathcal{Y} \to \mathcal{D}(\mathcal{X})$ assigns to each history-current output pair $(H, Y) \in (\mathcal{H}, \mathcal{Y})$ a conditional distribution $p_e(\cdot \mid H, Y)$ over the next observation,
- $p_{X_0} \in \mathcal{D}(\mathcal{X})$ specifies the distribution of the initial observation $X_0$.

---

[1]We assume that all measurable spaces $(\mathcal{X}, \mathcal{Y}, \mathcal{Z}, \dots)$ are standard Borel (Borel $\sigma$-algebras of Polish spaces), such that conditional probability kernels $p(\cdot \mid h, y)$ and $q(\cdot \mid z, x)$ exist and are measurable.

**Definition 3.4** (Learner). A *learner* relative to interface $(\mathcal{X}, \mathcal{Y})$ is a tuple $(\mathcal{Z}, f, u, u', p_{Z_0})$, where:

- a measurable space $\mathcal{Z}$, called the *learner state space*;
- a *prediction function* $f : \mathcal{Z} \times \mathcal{X} \to \mathcal{D}(\mathcal{Y})$, giving conditional distributions $q_f(\cdot \mid z, x)$;
- a *learning-mode state update function* $u : \mathcal{Z} \times \mathcal{X} \times \mathcal{Y} \to \mathcal{D}(\mathcal{Z})$;
- an *inference-mode state update function* $u' : \mathcal{Z} \times \mathcal{X} \times \mathcal{Y} \to \mathcal{D}(\mathcal{Z})$;
- an *initial learner state distribution* $p_{Z_0} \in \mathcal{D}(\mathcal{Z})$.

Although a learner's entire state could, in principle, be updated by a single function (as in work by Dong et al., 2022 and Kumar et al., 2025), we distinguish between two update functions to capture different modes of learner evolution. During interaction with the environment, the learning-mode update $u$ governs the evolution of the entire state, including predictive parameters and auxiliary components, such as replay buffers. In contrast, the inference-mode update $u'$ allows auxiliary components of the state, such as buffers or counters, to evolve while keeping the predictive parameters fixed. This distinction allows an observer to analyse the learner's behaviour in both dynamic conditions, where beliefs are updated over time, and in static conditions, where beliefs remain fixed.

**The interaction process.** The interaction between a learner and an environment defines a joint stochastic process over observations $X \in \mathcal{X}$, outputs $Y \in \mathcal{Y}$, and states $Z \in \mathcal{Z}$.

**Definition 3.5** (Interaction Process). The *(learning) interaction process* between an environment $(e, p_{X_0})$ and a learner $(\mathcal{Z}, f, u, u', p_{Z_0})$ relative to an interface $(\mathcal{X}, \mathcal{Y})$ is defined by:

$$
\begin{aligned}
\text{Initialisation:} \quad & Z_0 \sim p_{Z_0}, \quad X_0 \sim p_{X_0}, \\
\text{Interaction:} \quad & Y_t \sim q_f(\cdot \mid Z_{t-1}, X_{t-1}), && \text{(learner samples output)} \\
& X_t \sim p_e(\cdot \mid H_{0:t-1}, Y_t), && \text{(environment samples current observation)} \\
& Z_t \sim u(Z_{t-1}, X_t, Y_t). && \text{(learner updates state)}
\end{aligned}
$$

This interaction process generates the stochastic processes $X_t$ $(t \geq 0)$, $Y_t$ $(t > 0)$, and $Z_t$ $(t \geq 0)$.

## 3.2 PREDICTIVE DISTRIBUTIONS

At any time during learning, the learner's state encodes expectations about how future interactions will unfold. We can make these expectations explicit by asking the learner to *internally simulate the future*. Rather than waiting for new observations, we let the learner "roll forward" in inference mode using update $u'$ to project how the sequence of inputs and outputs may continue. This is very similar to placing a learner into inference mode, where it projects how the sequence of outputs might continue.

We refer to such a simulation as an *induced future* or *predictive distribution*. It represents a hypothetical rollout of the learner's predictive model under its current beliefs, independent from the environment. To distinguish this hypothetical evolution from real learning progression, we introduce a future time index $s \in \mathbb{N}$ and denote steps within the simulated trajectory as superscripts on $X$, $Y$, and $Z$.

During the rollout, the learner generates targets from its own predictive distribution, while any components not represented by the learner are *borrowed* from the environment. Formally, given a history $H_{0:t}$ and state $Z_t$, the predictive distribution evolves according to

$$Y^s \sim q_f(\cdot \mid Z_t^{s-1}, X^{s-1}), \quad X^s \sim q_e(\cdot \mid H_{0:s-1}, Y^s), \quad Z_t^s \sim u'(Z_t^{s-1}, X^s, Y^s), \tag{3}$$

where $q_e$ is a *hybrid distribution* that treats the learner's predictions as targets while borrowing components from the environment as needed, and $s = (t+1, t+2, \dots)$. Progression along the futures index $s$ provides a principled means to analyse the learner independently of new observations. We emphasise that this futures distribution is entirely isolated from the interaction process.

We refer to an element of future histories as $H \in \mathcal{H}$, which is a sequence of future observation-output pairs $(X^i, Y^i)$ induced by the learner $H^{t+1:\infty} = ((X^{t+1}, Y^{t+1}), (X^{t+2}, Y^{t+2}), \dots)$.

**Definition 3.6** (Predictive Distributions). The *predictive distribution* of the learner at state $Z_t$ with realised history $H_{0:t}$ is the joint distribution obtained by rolling out the inference-mode process in (3):

$$q(H^{t+1:\infty} \mid Z_t, H_{0:t}). \tag{4}$$

Thus, an induced future is a probability distribution over entire infinite sequences of future inputs and outputs $(\mathcal{X} \times \mathcal{Y})^{\mathbb{N}}$. Because the learner's state $Z_t$ and subsequent observations are random, the predictive distribution at each step $q(\cdot \mid Z_t, H_{0:t})$ is a random variable taking values in $\mathcal{D}((\mathcal{X} \times \mathcal{Y})^{\mathbb{N}})$.

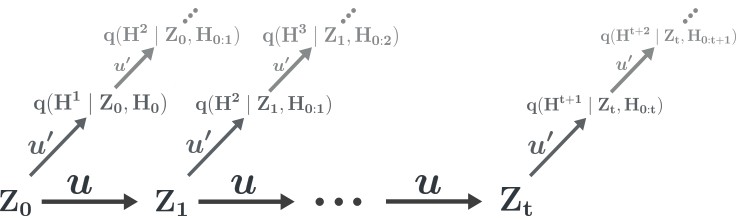

Figure 1: **State evolution and predictive distributions.** The learner's internal state $Z_t$ evolves through training updates $u$ given $(Z_{t-1}, X_t, Y_t)$. Each state induces a predictive distribution $q(H^{t+1:\infty} \mid Z_{t-1}, H_{0:t})$, which is updated in inference mode $u'$. This separation illustrates how each state encodes retained capabilities, and how training vs. introspective updates interact.

**Predictive-Bayesian perspective.** This construction motivates a *predictive perspective of learning*, inspired by predictive Bayesianism (Fortini & Petrone, 2019; 2025; Fong et al., 2023). Here, the learner's state $Z_t$ is characterised by its predictive distribution over futures. This perspective offers several advantages: it allows predictive statements to be *validated* against realised outcomes (Fong et al., 2023) and provides an interpretable representation of the learner's knowledge at any time – this is valuable in deep learning because parameters are not interpretable. By tracking the evolution of the predictive distribution, we yield a general formulation of learning that allows us to define forgetting.

### 3.3 LEARNING AS A STOCHASTIC PROCESS

The stochasticity of the environment induces stochasticity in the learner's state evolution; as the learner interacts with the environment, the sequence $\{Z_t\}$ forms a stochastic process. Since each state $Z_t$ defines a predictive distribution, the sequence of predictive distributions $\{q(H^{t+1:\infty} \mid Z_t, H_{0:t})\}$ also forms a stochastic process. Thus, the distribution over learner states at time $t$ defines a distribution over predictive distributions. This is illustrated in Figure 1.

**Interpretation across learning paradigms.** The abstract variables $(X_t, Y_t, Z_t)$ admit natural interpretations across learning settings. In supervised learning, $X_t$ denotes (current input, previous target) pairs and $Y_t$ the learner's output in response to the previous input; in RL, $X_t$ contains states and rewards while $Y_t$ are actions; in generative modelling, $Y_t$ directly models $X_t$. Across these paradigms, the learner state $Z_t$ encapsulates all contents of the learner (including parameters, latent variables, buffers, etc.). Distinct modes of operation correspond to different update rules: in training (update rule $u$), both the learner's belief and auxiliary structures may adapt, whereas an inference update ($u'$) leaves its beliefs fixed while transient components continue to evolve.

For example, in a supervised classification setting, $X_t$ consists of a (current input, previous target) pair, $Y_t$ is the label predicted by the learner given $X_{t-1}$, and $Z_t$ may represent parameters and momenta. The learning-mode update $u$ performs gradient steps on a loss that compares the observed and predicted labels. The inference-mode update $u'$ does not change the parameters and momenta.

Thus, while the same symbols are used throughout, their interpretation shifts depending on the paradigm. What remains consistent is the structural principle: the learner generates $Y_t$ conditioned on $(Z_{t-1}, X_{t-1})$, the environment supplies the observation $X_t$, and the state $Z_t$ evolves via $u$. This defines a single stochastic-process formalism that encompasses general learning processes.

## 4 FORGETTING

We consider an observer monitoring the interaction between a learner and its environment. The observer evaluates how the learner's capabilities evolve over time, with particular emphasis on quantifying the extent to which the learner forgets during the interaction.

### 4.1 CHARACTERISING FORGETTING

Before formalising forgetting, we first outline desiderata that any valid notion of forgetting should satisfy. These desiderata are motivated by thought experiments, detailed in §C.

*Desideratum* 4.1. A forgetting measure should quantify the loss of learned information over time.

Forgetting is distinct from measurable success on a task, such as accuracy or cumulative reward. It is a property of the learner itself, independent of the environment. A learner can maintain outdated or incorrect beliefs yet still perform well, or lose relevant knowledge without an immediate change in task outcomes. Conventional metrics, including backward transfer, may fail to capture these underlying changes in knowledge, highlighting the need to disentangle forgetting from task performance.

*Desideratum* 4.2. A characterisation of forgetting must not conflate forgetting with the correctness of outputs or with justified updates that change beliefs.

When a learner incorporates new observations, its beliefs (and therefore its state) will change. *A change in belief does not necessarily imply anything is forgotten.* Therefore, conceptualisations based on changes to beliefs (or parameters) can misidentify information-preserving updates as forgetting.

*Desideratum* 4.3. Forgetting should characterise the learner's loss of prior information and capabilities, not just the retention of previously observed data.

Forgetting encompasses the loss of general capabilities, not only individual observations. For example, a learner may fail to generalise to unobserved examples that it previously handled correctly. Conceptualisations of forgetting that prioritise memorisation overlook this broader notion.

*Desideratum* 4.4. Forgetting is a property of the learner, not of the environment in which it operates.

An environment cannot forget; however, it can influence the rate or magnitude of forgetting. These desiderata provide a principled foundation for developing and evaluating conceptualisations of forgetting. The thought experiments in §C justify the desiderata and formalism developed below.

## 4.2 FORGETTING

Learning is a process of both *absorption and loss*. Each new observation provides the learner with new information while simultaneously displacing or overwriting previously learned information. Therefore, every update reshapes what the learner can represent and, consequently, what it may forget.

If a learner's behaviour changes after observing information that it has already incorporated from previous observations, this cannot reflect the acquisition of new information. Therefore, it indicates a loss of prior knowledge. The learner's expectations are formalised by its predictive distribution; consequently, simulating training updates on samples drawn from this predictive distribution provides a principled perspective on the learner's likelihood of forgetting at any point in time.

Recall that the learner maintains a state that evolves recursively:

$$Z_t \sim u(Z_{t-1}, X_t, Y_t). \tag{5}$$

At each time $t$, the state $Z_t$ induces a predictive distribution $q(H^{t+1:\infty} \mid Z_t, H_{0:t})$, which characterises both the learner's current predictive capabilities and its expectations about futures.

We therefore define forgetting in terms of *induced futures*, rather than the learner's state. Different states–whether parametric, non-parametric, or otherwise–may induce identical futures:

$$q(H^{t+1:\infty} \mid Z_t, H_{0:t}) = q(H^{t+1:\infty} \mid Z_t', H_{0:t}), \tag{6}$$

where $Z_t \neq Z_t'$. Furthermore, predictions about already-seen observations may shift without indicating that forgetting has occurred. By grounding our definition in induced futures, we ensure that updates are evaluated with respect to the learner's *expected predictions*, thereby distinguishing between constructive and destructive adaptation. This satisfies Desideratum 4.2 and Desideratum 4.3.

**Consistency as non-forgetting.** Intuitively, a learner is *unforgetful* if, in expectation, its predictive distribution is invariant to updates on targets that are consistent with its own predictions.

Formally, let $q(H^{t+1:\infty} \mid Z_{t-1}, H_{0:t-1})$ denote the predictive distribution before updates on targets consistent with the learner's expectations, and let $q(H^{t+1:\infty} \mid Z_t, H_{0:t})$ denote the distribution after updating on targets consistent with the learner's expectations. Non-forgetting requires:

$$q(H^{t+1:\infty} \mid Z_{t-1}, H_{0:t-1}) = \mathbb{E}_{Y_t, X_t, Z_t} \left[ q(H^{t+1:\infty} \mid Z_t, H_{0:t}) \right], \tag{7}$$

where $Y_t \sim q_f(\cdot \mid Z_{t-1}, X_{t-1})$, $X_t \sim q_e(\cdot \mid H_{0:t-1}, Y_t)$, and $Z_t \sim u(\cdot \mid Z_{t-1}, X_t, Y_t)$.

In these updates, $X_t$ is sampled from the same hybrid distribution $q_e$ introduced in §3.2. Updates are performed on these learner-consistent targets to ensure a separation between forgetting and backward

transfer. In this formulation, the expectation in (7) is taken over both the stochasticity of the learner's target generation and the environmental inputs, yielding a description of predictive consistency.

This notion extends to multiple updates. Non-forgetting requires that predictive distributions remain compatible with those induced after updates. Formally, predictive distributions must be recoverable by marginalising over all $k$-step interaction paths, yielding the *generalised consistency condition*.

**Definition 4.5** (Consistency Condition). For $k \geq 1$, a learner is $k$-step consistent *if and only if*

$$q_k^*(H^{t+k:\infty} \mid Z_{t-1}, H_{0:t-1}) = \mathbb{E}_{X_{t:t'}, Y_{t:t'}, Z_{t:t'}} \left[ q(H^{t+k:\infty} \mid Z_{t'}, H_{0:t'}) \right], \qquad (8)$$

where $t' = t + k - 1$, and for $i = t, \ldots, t'$ the expectation is taken over $X_i \sim q_e(\cdot \mid H_{0:i-1}, Y_i)$, $Y_i \sim q_f(\cdot \mid Z_{i-1}, X_{i-1})$, $Z_i \sim u(\cdot \mid Z_{i-1}, X_i, Y_i)$.

Definition 4.5 shows why replay is often essential. When the update $u(Z_{t-1}, X_t, Y_t)$ depends on the history $H_{0:t-1}$, then the consistency condition requires access to past data during updates. Replay mechanisms provide this access, offering a clear mathematical justification for their role (see §B.3).

**Propensity to forget.** In practice, most learning algorithms do forget. To quantify *how much* the learner is likely to forget at any time, we introduce a concrete measure grounded in our formalism. When the consistency condition is violated, the learner's updated predictive distribution diverges from its initial predictive distribution. Measuring this divergence yields a natural notion of a learner's *propensity to forget*. This operationalises our conceptual definition, allowing us to validate the formalism by ensuring that the measure aligns with intuitive expectations about forgetting in practice.

**Definition 4.6** (Propensity to Forget). The $k$-step propensity to forget incurred at time $t$, using a suitable divergence measure $D(\cdot \| \cdot)$, is given by

$$\Gamma_k(t) := D\left( q(H^{t+k:\infty} \mid Z_{t-1}, H_{0:t-1}) \,\|\, q_k^*(H^{t+k:\infty} \mid Z_{t-1}, H_{0:t-1}) \right). \qquad (9)$$

> **Takeaway 1.** *Forgetting occurs when the consistency condition is violated; the predictive distribution after $k$ updates is no longer recoverable from those before the updates.*

**Scope and boundary of validity.** Our formalism applies whenever the learner's predictive distribution accurately represents the learner's state, $Z_t \mapsto q(H^{t+1:\infty} \mid Z_t, H_{0:t})$. Only information used to generate predictions contributes; state components that do not influence predictions (e.g., unused buffer entries) are excluded. Typically, the predictive distribution reflects the state, but this may not be the case during transitory phases such as buffer reinitialisation, target-network lag, or other mechanisms that temporarily decouple the state from predictions. In these intervals, forgetting is undefined, not because the formalism fails, but because the learner temporarily lacks a predictive model of its behaviour. Some algorithms may never produce a predictive mapping and thus fall outside the scope of this formalism. In most cases, however, the predictive distribution is representative of the state.

## 5 EMPIRICAL ANALYSIS

Our theoretical account provides a general conceptualisation of forgetting. To illustrate its utility, we empirically study Definition 4.6 across multiple environments and learning algorithms.

### 5.1 UNFORGETFUL LEARNERS

Exact Bayesian learners are unforgetful because they satisfy the $k$-step self-consistency condition: marginalising the posterior after a hypothetical future observation recovers the current posterior,

$$p(\theta \mid X_{1:t}) = \int p(\theta \mid X_{1:t+1}) p(X_{t+1} \mid X_{1:t}) \, dX_{t+1}. \qquad (10)$$

Conditioning on a hypothetical future observation and then marginalising over its prior predictive distribution returns the same belief as not conditioning at all; conditioning and marginalising commute. In exchangeable settings, this self-consistency further implies permutation-invariance. Let $X_{1:t}$ denote the observations up to time $t$, and $\theta$ the model parameters. The Bayesian posterior is

$$p(\theta \mid X_{1:t}) \propto p(\theta) \prod_{i=1}^{n} p(X_i \mid \theta). \qquad (11)$$

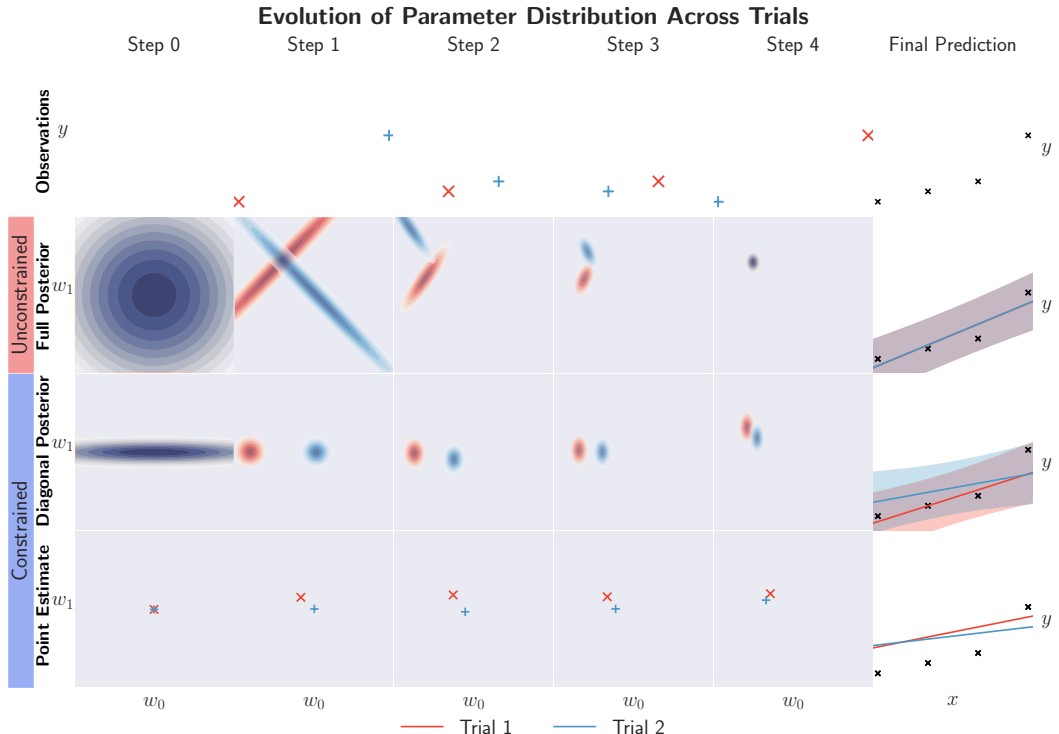

Figure 2: **Self-consistent learners do not forget.** Axes showing **observations** $(x, y)$ are shaded white, and **parameter** axes $(w_0, w_1)$ are shaded grey. *Top row:* The same four observations are presented to a linear regression learner in different orders. *Second row:* A full Bayesian posterior represents a complete summary of all observations and satisfies the $k$-step consistency condition, implying that no forgetting has occurred. The permutation invariance of the learner in exchangeable settings follows from the commutativity of Bayesian updates. *Bottom rows:* Two constrained learners – a Gaussian variational posterior with diagonal covariance and a point estimate updated by gradient descent – violate self-consistency. Their updates alter induced futures; consequently, previously supported capabilities may be lost. The final column shows the resulting posterior predictive distributions.

Since multiplication commutes, the posterior is invariant to the order of observations:

$$p(\theta \mid X_{1:t}) = p(\theta \mid X_{\pi(1)}, \dots, X_{\pi(t)}) \quad \forall \text{ permutations } \pi. \tag{12}$$

Therefore, exact Bayesian updates are *permutation-invariant* in exchangeable settings.

Approximate learners, such as those using diagonal Gaussian variational inference or gradient-based point estimates, violate self-consistency. Their updates depend on the order of observations, and their beliefs cannot be recovered by marginalising over future predictions. Consequently, the learner's current predictive distribution may exclude previously supported predictions, which by Definition 4.5 constitutes forgetting. The contrast between exact and approximate learners is shown in Figure 2.

> **Takeaway 2.** *Parameter changes alone do not imply forgetting.*

## 5.2 Forgetting in Deep Learning

Across deep learning settings, forgetting is consistently non-zero, with both its magnitude and dynamics varying substantially (Figure 3). Even in i.i.d. settings, where forgetting is often overlooked, forgetting fluctuates throughout training, reflecting how neural networks continually update and overwrite their knowledge. Although absolute $\Gamma_k(t)$ values are domain-specific, the forgetting trajectory reveals how the learner's beliefs evolve over time. These results show that forgetting is functionally meaningful in all tasks, highlighting the importance of a general conceptualisation of forgetting. This empirical observation motivates our paper title, "Forgetting is Everywhere".

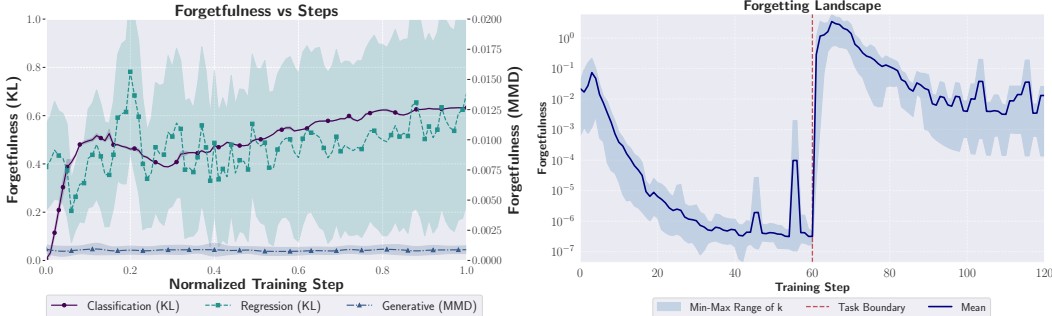

Figure 3: **Forgetting occurs across all deep learning scenarios.** *Left:* Forgetfulness dynamics of a shallow neural network trained on regression, classification, and generative modelling tasks. The solid line shows the $k$-step forgetfulness (where $k$ varies from 1 to 40) over the normalised training step. Regression and classification tasks use KL divergence, while the generative task uses the maximum mean discrepancy (MMD). Forgetting dynamics vary throughout training, even without any distribution shift. *Right:* A class-incremental learning example using a single-layer neural network on a two-moons classification task. We show the $k$-step forgetting profile over four seeds, with the shaded area indicating the spread of $\Gamma_k(t)$ over $k$ from 1 to 40. The plot illustrates the abrupt increase in forgetting at the task boundary. See §F for details on the experimental implementation.

## 5.3 APPROXIMATE LEARNERS CAN BENEFIT FROM FORGETTING

At each update, an approximation-based learner incorporates new information from current observations while discarding parts of its existing state (§4.2). Because approximate updates yield imperfect representations, a learner's performance depends on striking a balance between adapting to new information and retaining useful prior information.

To study this effect, we investigate how modifications to the learner influence the propensity to forget. Across experiments, a consistent pattern emerges for approximate learners: *a moderate amount of forgetting improves learning efficiency*. Here, we quantify training efficiency using the inverse of the normalised area under the training loss curve, a practical proxy for learning speed and convergence quality. Empirically, the forgetting-efficiency relationship shows an "elbow" (Figure 4), indicating that optimal training efficiency occurs at a non-zero level of forgetting. This suggests that effective approximate learners utilise forgetting as a mechanism for adaptive and efficient learning.

> **Takeaway 3.** *Forgetting is ubiquitous in deep learning. The trade-off between training efficiency and forgetting determines the optimal amount to forget–in deep learning, this is rarely zero.*

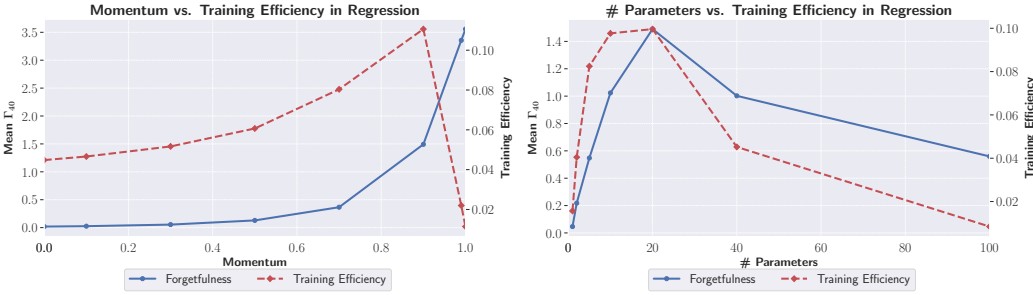

Figure 4: **Approximate learners can benefit from non-zero forgetting.** We analyse how training efficiency and forgetfulness co-vary across learning algorithms. Training efficiency is measured as the inverse of the normalised area under the training loss curve, an approximate but informative proxy for learning speed and convergence quality in the setting we study. Forgetfulness is quantified as the mean 40-step propensity to forget, $\Gamma_{40}(t)$, across training steps. *Left:* In a regression task, varying stochastic gradient descent's momentum parameter shows that higher momentum increases forgetfulness, with maximum training efficiency at $0.9$ momentum. *Right:* Varying model size shows that maximum efficiency occurs at 20 parameters. In both cases, the most efficient learners exhibit some forgetting–too little slows adaptation, too much destabilises learning–highlighting a fundamental trade-off.

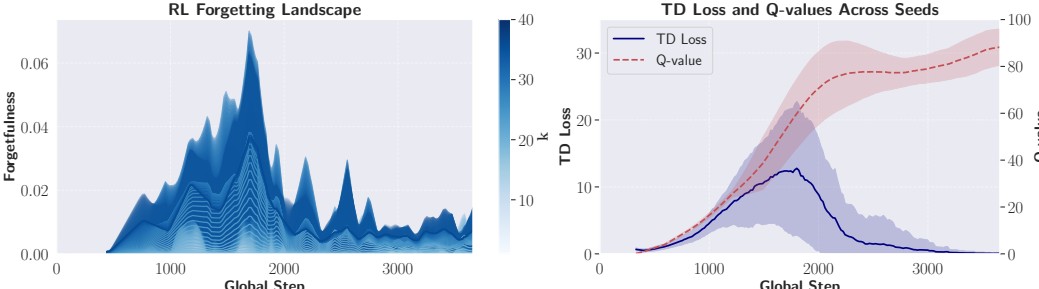

Figure 5: **Forgetting in RL reflects active management of the information acquisition-retention trade-off.** We show TD loss, Q-value evaluation, and the forgetting profile for a DQN learner trained on cartpole across ten seeds. Early in training, TD loss is low; it rises as the agent acquires new information, then decreases once knowledge is integrated, plateauing as information acquisition slows. The forgetting curve follows this trajectory, highlighting that forgetting old information is a deliberate mechanism for balancing knowledge acquisition with knowledge retention.

### 5.4 IMPLICATIONS OF DISTRIBUTION SHIFT

Distribution shift and stochasticity strongly influence forgetting dynamics. In i.i.d. environments, the interaction process is stationary and the predictive distributions are stable. When learner hyperparameters are well-tuned and training conditions are stable, the learner effectively balances adaptation and retention. In such settings, training efficiency is high because the learner's updates operate under consistent conditions, and the underlying data distribution provides consistent feedback for improvement.

In CL, abrupt shifts in the observation distribution cause discontinuous changes in state, $Z_t$, and the predictive distribution. Consequently, the *magnitude of consistency violation abruptly increases at task boundaries* (Figure 3) as the learner must rapidly adapt to a new task.

All learning involves balancing the integration of new information with the retention of current knowledge. RL presents this challenge in an extreme form. Here, the learner's policy influences future observations, inducing continual non-stationarity. In DQN, for example, as the agent experiences new transitions, the TD loss rises because the agent incorporates new information (Figure 5). As the agent consolidates this information, the TD loss declines and the rate of information acquisition plateaus. The forgetting curve follows the TD loss because forgetting information is the mechanism by which the agent manages this process, demonstrating that forgetting is an essential component of RL.

> **Takeaway 4.** *Forgetting is an integral component of learning: effective learning requires selectively forgetting outdated knowledge to integrate new information.*

## 6 CONCLUSION

In this work, we introduced a general, algorithm- and task-agnostic formulation of forgetting, describing it as the *temporal inconsistency* of a learner's predictive distribution. To our knowledge, this is the *first generalised definition of forgetting*. Unlike previous definitions, our approach encompasses generalisation forgetting, disentangles forgetting from backward transfer, and separates forgetting from parameter updates. This shows that learners *can adapt without forgetting* (§5.1). We also introduced the *propensity to forget* as an operational measure, allowing empirical validation of our definition.

Our empirical analysis across diverse learning algorithms and task settings shows that *forgetting is pervasive in deep learning* and shaped by interactions between the learner and the environment. Interestingly, optimal training efficiency does not always correspond to minimal forgetting; in some cases, an *intermediate amount of forgetting maximises efficiency*, highlighting the importance of considering forgetting when designing learning algorithms.

Overall, our work reframes forgetting as a *fundamental property of learning dynamics* rather than a failure mode limited to continual or non-stationary regimes. We hope our work provides a clear conceptual basis for understanding how a learner's capabilities emerge, persist, and deteriorate, guiding the design of algorithms that can adapt while retaining previously acquired knowledge.

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

## APPENDIX

The appendix is organised as follows: §A defines the notation, §B expands on the theory, §C gives illustrative thought experiments, §D provides additional details regarding empirical implementation, §E reports additional results, and §F details our experiments.

# A NOTATION

We first provide a table summarising the notation used throughout the paper.

Table 1: A summary of notation.

| Symbol | Description | Domain / Type |
|---|---|---|
| **Interface-related** | | |
| $(\mathcal{X}, \mathcal{Y})$ | Environment–learner interface | Pair of measurable spaces |
| $\mathcal{X}$ | Observation space | Measurable space |
| $\mathcal{Y}$ | Prediction space | Measurable space |
| $\mathcal{H}$ | Set of histories | $\bigcup_{t=0}^{\infty} (\mathcal{X} \times \mathcal{Y})^t$ |
| $X_t$ | Observation at time $t$ | Element of $\mathcal{X}$ |
| $Y_t$ | Output at time $t$ | Element of $\mathcal{Y}$ |
| $H_{0:t}$ | History up to time $t$ | Sequence $((X_1, Y_1), \ldots, (X_t, Y_t))$ |
| **Environment-related** | | |
| $(e, p_{X_1})$ | Environment | $p_{X_1} \in \mathcal{D}(\mathcal{X}),\ e : \mathcal{H} \to \mathcal{D}(\mathcal{X})$ |
| $p_e(\cdot \mid h, y)$ | Environment distribution given history $h$ and output $y$ | $\mathcal{D}(\mathcal{X})$ |
| $q_e(\cdot \mid h, y)$ | Learner-sampled hybrid input distribution given history $h$ and output $y$ | $\mathcal{D}(\mathcal{X})$ |
| **Learner-related** | | |
| $(\mathcal{Z}, f, u, u', p_{Z_0})$ | Learner | Tuple |
| $\mathcal{Z}$ | Learner state space | Space |
| $Z_t$ | Learner state at time $t$ | Element of $\mathcal{Z}$ |
| $Z_t^s$ | Learner state in inference mode at a futures time $s$ | Element of $\mathcal{Z}$ |
| $p_{Z_0}$ | Initial learner state distribution | $\mathcal{D}(\mathcal{Z})$ |
| $f$ | Prediction function | $\mathcal{Z} \times \mathcal{X} \to \mathcal{D}(\mathcal{Y})$ |
| $u$ | Learning-mode state update | $\mathcal{Z} \times \mathcal{X} \times \mathcal{Y} \to \mathcal{D}(\mathcal{Z})$ |
| $u'$ | Inference-mode state update | $\mathcal{Z} \times \mathcal{X} \times \mathcal{Y} \to \mathcal{D}(\mathcal{Z})$ |
| $q_f(\cdot \mid z, x)$ | Predictive distribution | $\mathcal{D}(\mathcal{Y})$ |
| **Induced futures** | | |
| $q(H^{t+1:\infty} \mid Z_t, H_{0:t})$ | Induced future distribution | $\mathcal{D}((\mathcal{X} \times \mathcal{Y})^{\mathbb{N}})$ |
| $X^s$ | Future observation at future time $s$ | Element of $\mathcal{X}$ |
| $Y^s$ | Future output at future time $s$ | Element of $\mathcal{Y}$ |
| $H^{t+1:\infty}$ | Sampled induced future at time $t$ | Sequence in $((X^{t+1}, Y^{t+1}), (X^{t+2}, Y^{t+2}), \ldots)$ |

## A.1 FRAMEWORK GENERALITY

We now provide explicit examples of different realisations of the abstract variables introduced in §3.

Table 2: This table provides specific examples of values for the abstract variables: $X_t, Y_t$ and $Z_t$ across different common machine learning settings.

| Setting | $X_t$ (observation) | $Y_t$ (prediction) | $Z_t$ (learner state) |
|---|---|---|---|
| Regression | Current input, previous target pair $(x_t,\ y_{t-1})$ | Predicted target $\hat{y}_{t-1}$ | Parameters (e.g., weights, optimiser state) |
| Classification | Current input, previous label pair $(x_t,\ \ell_{t-1})$ | Predicted class label $\hat{\ell}_{t-1}$ | Classifier parameters, latent state (e.g., neural network weights, momenta) |
| Generative modelling | Data to generate $x_t$ | Attempted generations of $x_t$ | Generative model parameters, latent variables, buffers |
| Bayesian models | Data (possibly $(x_t, y_{t-1})$ pairs) | Posterior predictive sample $\hat{y}_{t-1}$ | Posterior / variational parameters, latent variables |
| Reinforcement learning | State–reward $(s, r)$ or transition $(s, a, r, s')$ | Action $a$ (policy output) or predictive model of next state | Policy/value parameters, transition/reward models, replay buffer, auxiliary structures |

–appendices continue on next page–

## B   THEORETICAL RESULTS

Recall, the learner's state is updated according to

$$Z_t \sim u(Z_{t-1}, X_t, Y_t). \tag{13}$$

Before observing $X_t, Y_t$ and updating to $Z_t$, we have a futures distribution

$$q(H^{t+1:\infty} \mid Z_{t-1}, H_{0:t-1}). \tag{14}$$

### B.1   ONE-STEP CONSISTENCY CONDITION

After the update, a non-forgetting learner satisfies the following conditional independence rules:

$$H^{t+1:\infty} \perp\!\!\!\perp Z_{t-1} \mid Z_t, H_{0:t}, \qquad Y_t \perp\!\!\!\perp H_{0:t-2}, Y_{t-1} \mid Z_{t-1}, X_{t-1},$$
$$Z_t \perp\!\!\!\perp H_{0:t-1} \mid Z_{t-1}, X_t, Y_t. \tag{15}$$

Furthermore, the environment admits the following independence:

$$X_t \perp\!\!\!\perp Z_{t-1}. \tag{16}$$

The one-step consistency operator is obtained by marginalising over observations $X_t, Y_t$ and updated state $Z_t$:

$$q^*(H^{t+1:\infty} \mid Z_{t-1}, H_{0:t-1}) = \int q(H^{t+1:\infty}, X_t, Y_t, Z_t \mid Z_{t-1}, H_{0:t-1}) \, \mathrm{d}X_t \, \mathrm{d}Y_t \, \mathrm{d}Z_t. \tag{17}$$

This can factorised, with the conditional independence rules in (15) and (16) applied, to derive the following:

$$\int q(X_t, Y_t, Z_t, H^{t+1:\infty} \mid Z_{t-1}, H_{0:t-1}) \, \mathrm{d}X_t \, \mathrm{d}Y_t \, \mathrm{d}Z_t$$
$$= \int q(H^{t+1:\infty} \mid Z_t, H_{0:t}) \, u(Z_t \mid Z_{t-1}, X_t, Y_t)$$
$$q_e(X_t \mid H_{0:t-1}, Y_t) \, q_f(Y_t \mid Z_{t-1}, X_{t-1}) \, \mathrm{d}X_t \, \mathrm{d}Y_t \, \mathrm{d}Z_t. \tag{18}$$

which is the 1-step consistency condition from (7):

$$q(H^{t+1:\infty} \mid Z_{t-1}, H_{0:t-1}) = \mathbb{E}_{X_t, Y_t, Z_t} \left[ q(H^{t+1:\infty} \mid Z_t, H_{0:t}) \right], \tag{19}$$

where

$$X_t \sim q_e(\cdot \mid H_{0:t-1}, Y_t), \quad Y_t \sim q_f(\cdot \mid Z_{t-1}, X_{t-1}), \quad Z_t \sim u(\cdot \mid Z_{t-1}, X_t, Y_t).$$

### B.2   K-STEP CONSISTENCY CONDITION

We can apply the same reasoning to derive the $k$-step consistency condition. If we denote $t'$ as $t' = t + k - 1$, we get:

$$q_k^*(H^{t+k:\infty} \mid Z_{t-1}, H_{0:t-1})$$
$$= \int q(X_{t:t'}, Y_{t:t'}, Z_{t:t'}, H^{t+k:\infty} \mid Z_{t-1}, H_{0:t-1}) \, \mathrm{d}X_{t:t'} \, \mathrm{d}Y_{t:t'} \, \mathrm{d}Z_{t:t'}$$
$$= \int q(H^{t+k:\infty} \mid Z_{t'}, H_{0:t'}) \cdot \prod_{s=t}^{t'} \left[ u(Z_s \mid Z_{s-1}, X_s, Y_s) \right.$$
$$\left. q_e(X_s \mid H_{0:s-1}, Y_s) q_f(Y_s \mid Z_{s-1}, X_{s-1}) \right] \mathrm{d}X_{t:t'} \, \mathrm{d}Y_{t:t'} \, \mathrm{d}Z_{t:t'}, \tag{20}$$

which is the $k$-step consistency condition from Definition 4.5:

$$q_k^*(H^{t+k:\infty} \mid Z_{t-1}, H_{0:t-1}) = \mathbb{E}_{X_{t:t'}, Y_{t:t'}, Z_{t:t'}} \left[ q(H^{t+k:\infty} \mid Z_{t'}, H_{t'}) \right]. \tag{21}$$

where, for $i = t, \ldots, t'$

$$X_i \sim q_e(\cdot \mid H_{0:i-1}, Y_i), \quad Y_i \sim q_f(\cdot \mid Z_{i-1}, X_{i-1}), \quad Z_i \sim u(\cdot \mid Z_{i-1}, X_i, Y_i).$$

### B.3 THE THEORETICAL NEED FOR REPLAY

The above derivations make explicit the role of the learner's state update function $u(Z_{t-1}, X_t, Y_t)$ in ensuring the consistency of the induced futures. In particular, the one-step and $k$-step consistency conditions rely on the assumption that the update function is such that $Z_t$ is conditionally independent of the whole history $H_{0:t-1}$ given $(Z_{t-1}, X_t, Y_t)$.

However, in practice, many learning algorithms violate this condition of conditional independence, and the update function explicitly depends on the history. When this dependence exists, correctly performing consistent updates depends on access to past data. Replay mechanisms provide an empirical solution: by placing past observations into the current observation, the learner effectively restores the conditional independence assumed by the theoretical consistency conditions.

Thus, from a theoretical perspective, the need for replay arises naturally whenever the state update $u(Z_{t-1}, X_t, Y_t)$ is insufficient at capturing all historical dependencies. Replay promotes consistent induced futures by explicitly placing much of the history into the current observation and, as a consequence, prevents forgetting that would otherwise occur.

## C    THOUGHT EXPERIMENTS

We present a series of thought experiments designed to stress-test potential definitions of forgetting in general learning systems. Each scenario illustrates a conceptual edge case that any robust definition of forgetting should naturally handle. For each scenario, we provide a *consistency verdict*, indicating whether our framework classifies the learner to be forgetting or not, and explain the intuition behind this judgement.

**Scenario C.1 *(The degenerate learner.)*.**  Consider a learner that never updates its beliefs–its state is fixed for all time. Such a learner never changes its state or acquires new information and is thus degenerate. Does this learner ever forget? One might argue that forgetting is defined relative to the information available to the learner over time; however, if nothing is ever learned, how can anything ever be forgotten?

*Consistency verdict: no forgetting.* Since the state remains constant across any number of updates, the induced futures distributions are identical. A learner that never changes its state can never forget.

**Scenario C.2 *(0-bit and 5-bit stacks.)*.**  Consider two first-in, first-out (FIFO) stacks: one with capacity 0 bits, another with 5 bits. The 0-bit stack cannot store information; the 5-bit stack can.

*Consistency verdict: 0-bit does not forget, the 5-bit stack does.* The predictive distribution of the stack is always the returned bit of the address being accessed at each step. A 0-bit stack's state never changes, so its future remains constant. The 5-bit stack, however, overwrites past bits when it is full; its predictive distribution changes as bits are dropped, hence it forgets. Forgetting only exists in systems capable of learning.

**Scenario C.3 *(The hash map.)*.**  A hash map with infinite associative memory never overwrites existing entries. It represents a theoretical "perfect rememberer".

*Consistency verdict: no forgetting.* The predictive distributions of such a learner are a Dirac at the retrieved value (or null if unassigned), and the induced futures remain identical over time. Thus, the learner never forgets.

**Scenario C.4 *(The clock.)*.**  Consider a simple clock that increments its state deterministically with each tick. Does the clock forget past states? In one sense, yes: it overwrites its register and never recovers earlier times. In another sense, no: it never misrepresents the current time and updates consistently. This raises the question: must forgetting be tied to the loss of recoverable information, or only to deviations from self-consistent updates?

*Consistency verdict: no forgetting.* Under regular environmental updates, the clock advances its state (its notion of "time") by one unit per tick. However, under consistency updates, we propagate the learner's state forward using its own current beliefs rather than the actual environmental progression. Since the clock believes time remains fixed at its last known state, it will continue to predict the exact time indefinitely. Consequently, its futures remain invariant under consistency updates. The clock does not forget.

**Scenario C.5 *(The moody learner.)*.**  Suppose a learner only updates on even timesteps, ignoring all odd ones. Does inaction indicate perfect memory?

*Consistency verdict: no forgetting (on ignored steps).* When no update occurs, the state remains identical, so induced futures distributions are unchanged. Forgetting can only occur during state-altering updates.

**Scenario C.6 *(The function picker.)*.**  Suppose there are $L$ learner functions, and at each timestep, one function is selected uniformly at random, independent of both $t$ and the observed data.

*Consistency verdict: no forgetting.* In expectation, the induced futures after any number of consistent updates remain invariant. The randomness of selection does not imply loss of information about possible futures.

**Scenario C.7 *(The binary flipper.)*.**  Consider a binary classification model that flips its predictions at every timestep.

*Consistency verdict: no forgetting.* Similar to the clock, the consistency update believes time remains fixed at its last known state; thus, the learner does not flip its predictions, and its state remains fixed. Consequently, its futures remain invariant under consistency updates, and the flipper does not forget.

**Scenario C.8** *(Label permutation.)*. Take a perfectly trained binary classifier. Now, consider that we permute the mapping between logits and output labels without changing the parameters. On the one hand, the parameters remain the same; on the other, the outputs have changed significantly. Has such a model forgotten everything it knew just because its behaviour has changed, or has it not forgotten anything at all because it has the same states?

*Consistency verdict: no forgetting.* Similar to the clock, the consistency update believes time remains fixed at its last known state; thus, the learner does not flip its predictions during consistent updates, and its state and predictive distributions remain fixed. Consequently, its futures remain invariant under consistency updates, and the learner does not forget.

**Scenario C.9** *(Forgetting unseen but generalised inputs.)*. Consider a learner that is trained on the MNIST dataset. At time $t$, the model correctly classifies all instances of the digit "4" in the test set. After further training, it loses the ability to correctly classify some test examples. Can a model forget something that it has never directly encountered (such as specific test inputs) but has nonetheless learned to generalise to?

*Consistency verdict: this forgets.* Even though the test data were unseen, the learner's predictive distribution changes when its performance on those data changes. Thus, the learner has forgotten: forgetting occurs whenever the previously supported predictive capabilities vanish, regardless of whether the corresponding data have been observed.

**Scenario C.10** *(Even number checkers.)*. Suppose a model receives binary inputs sequentially and deterministically predicts a $1$ if it has observed an even number of 1s, and $0$ otherwise. Does it forget because its state changes over time?

*Consistency verdict: no forgetting.* The next $k$ observations may either be a $0$ or a $1$, drawn from any distribution. After $k$ updates, the learner's predictive distribution is deterministically either a $0$ or a $1$. Although the state continues to evolve across the induced futures, the expected predictive distribution remains the same across them. Therefore, no forgetting occurs.

**Scenario C.11** *(Surprising events.)*. Consider the scenario where a fair coin is flipped repeatedly. After 10 coin flips, we are in an unlikely situation where all flips have so far resulted in heads. If, on the next flip, we observe a tail, the learner may be surprised, even if it has remembered all previous results.

*Consistency verdict: no forgetting.* Under consistent updates, future updates are performed relative to the learner's current beliefs rather than the actual environment. This distinction is what prevents surprise from being mistaken for forgetting. Since the learner's predictive distribution already encodes its own uncertainty over future outcomes, an unexpected tail does not constitute loss of information; it is a valid observation under the learner's model. Consequently, the expected futures distribution before and after the $k$ updates remains identical, and no forgetting is detected.

**Scenario C.12** *(Bayesian optimisation.)*. Consider a Bayesian optimiser that initially observes high objective values, then later finds a lower minimum. Its predictive distribution must update in response to the new observation. However, no previously acquired evidence has been discarded. This is a knowledge *update*, not forgetting; naive comparisons of states may mistakenly classify this update as forgetting.

*Consistency verdict: no forgetting.* As in Scenario C.11, future updates are computed relative to the learner's current predictive distribution rather than the actual environment. Because updates reflect the learner's own beliefs, encountering a new minimum does not imply a loss of prior information. In expectation, the futures distributions before and after $k$ updates remain identical, indicating that the learner has not forgotten.

–appendices continue on next page–

# D EMPIRICAL CONSIDERATIONS

In this section, we detail the approach used to estimate the forgetfulness measure $\Gamma_k(t)$. $\Gamma_k(t)$ captures the extent to which the learner's induced predictive distribution at time $t$ remains consistent with those obtained after $k$ simulated updates. While this quantity cannot be computed exactly, it can be approximated using a particle-based Monte Carlo scheme. At a given time, we clone the learner into $M$ replicas and propagate each replica forward under $k$ steps of simulated interaction with its own induced futures. This yields a set of post-update models $\{Z_{t+k-1}^{(m)}\}_{m=1}^M$, whose mixture distribution provides an approximation to the consistency condition:

$$q_k^*(H^{t+k:\infty} \mid Z_{t-1}, H_{0:t-1}) \approx \frac{1}{M} \sum_{m=1}^M q(H^{t+k:\infty} \mid Z_{t+k-1}^{(m)}, H_{t+k-1}^{(m)}), \tag{22}$$

We compare the learner's distribution on a reference trajectory of simulated futures with this mixture distribution. The divergence between the two provides an empirical estimate of $\Gamma_k(t)$. This is detailed in Algorithm 1, provided below.

**Predictive distributions.** In practice, the predictive distribution $q_f(Y_t \mid Z_{t-1}, X_{t-1})$ is obtained directly from the model's output parameterisation. For classification tasks, this distribution is explicit: the network's logits define a categorical (or Bernoulli) distribution over discrete classes. A similar interpretation applies to generative models, where the predictive distribution corresponds to the model's generative distribution over data samples. For regression tasks, the predictive distribution is determined by the likelihood implied by the training loss. When optimising a mean-squared-error objective, for example, the corresponding likelihood is Gaussian, so the model's prediction corresponds to the mean of a normal distribution. The predictive variance can be estimated empirically from the residual error distribution on a held-out validation set. Thus, even though most practical neural networks output only point predictions, their underlying training objective defines an implicit predictive distribution. We use this to evaluate the induced futures and empirically quantify forgetting.

**Visualising the propensity to forget** Figure 6 compares a single-layer neural network's futures distribution at the current training step with the same model but after $40$ additional updates on futures data. This allows us to visualise the divergence between the two induced futures distributions. This difference quantifies the model's propensity to forget, as defined in Definition 4.6.

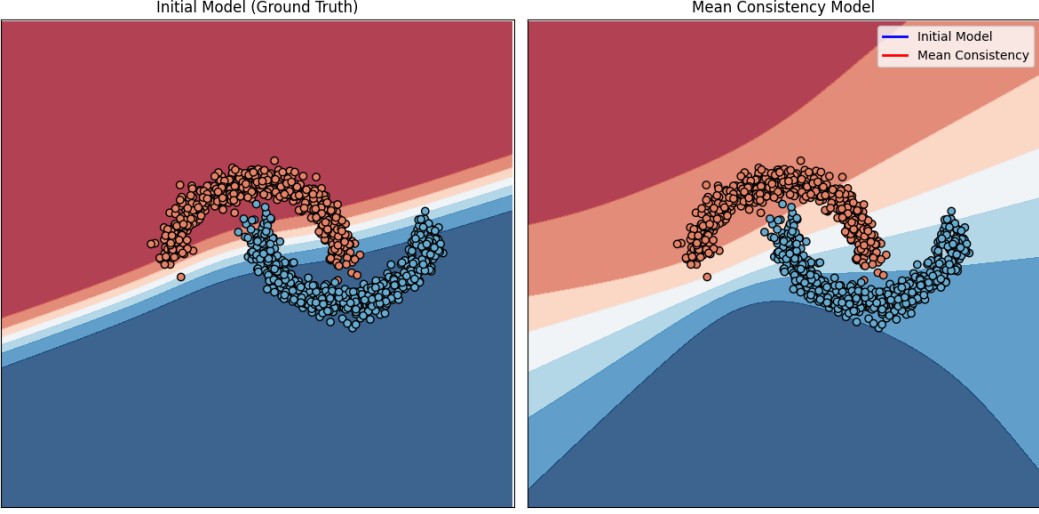

Figure 6: **Comparison of initial and 40-step induced futures.** Axes show the input space around the two-moon binary classification task. Example datapoints are overlaid and coloured by target class (red = 0, blue = 1). The background grid is shaded according to the classifier's logit values, from red (0) to blue (1), with white indicating uncertainty at 0.5. *Left:* The initial induced futures distribution $q(H^{t+k:\infty} \mid Z_{t-1}, H_{0:t-1})$ serves as the baseline for consistent updates. *Right:* The $k = 40$ induced futures distribution $q_k^*(H^{t+k:\infty} \mid Z_{t-1}, H_{0:t-1})$, showing how the predictive model is likely to evolve over 40 steps. The divergence in Definition 4.6 is computed between these two distributions.

## D.1 Supervised classification example

Consider a supervised classification setting with a neural network learner. To approximate future inputs $X_{t:\infty}$, we sample uniformly over the empirical distribution of inputs observed thus far. This is acquired by sampling $k$, $X$ values from a held-out validation set, $X_{t:t+k-1}$, and using the remaining inputs to evaluate predictive distributions, approximating $X_{t+k:\infty}$.

We first compute the predictive distribution of the current model, $Z_{t-1}$.

$$q(Y_{t+k:\infty} \mid Z_{t-1}, X_{t+k-1:\infty}), \tag{23}$$

which will act as the reference distribution in the KL divergence.

To form the $k$-step consistency forecast, we perform a Monte Carlo estimate: make $N$ independent particle copies of the learner and, for each particle $n$ and for $s = 1, \ldots, k$, we do the following:

$$
\begin{aligned}
\text{(sample target)} \quad & Y_{t+s}^{(n)} \sim q_f(\cdot \mid Z_{t+s-1}^{(n)}, X_{t+s-1}^{(n)}), \\
\text{(sample input)} \quad & X_{t+s}^{(n)} \sim q_e(\cdot \mid H_{0:t+s-1}, Y_{t+s}^{(n)}), \\
\text{(update state)} \quad & Z_{t+s}^{(n)} \sim u(\cdot \mid Z_{t+s-1}^{(n)}, X_{t+s}^{(n)}, Y_{t+s}^{(n)}).
\end{aligned} \tag{24}
$$

After $k$ steps, each particle yields a predictive distribution over the future inputs,

$$q_k^{(n)}(Y_{t+k:\infty} \mid Z_{t+k-1}^{(n)}, X_{t+k-1:\infty}^{(n)}). \tag{25}$$

We form the uniform mixture to acquire the consistency forecast,

$$q_k^*(Y_{t+k:\infty} \mid Z_{t+k-1}, X_{t+k-1:\infty}). \tag{26}$$

We can then compute the learner's propensity to forget:

$$\Gamma_k(t) = D_{\mathrm{KL}}\left(q(Y_{t+k:\infty} \mid Z_{t-1}, X_{t+k-1:\infty}) \,\|\, q_k^*(Y_{t+k:\infty} \mid Z_{t+k-1}, X_{t+k-1:\infty})\right). \tag{27}$$

This process is described in Algorithm 1.

---

**Algorithm 1** Computing the propensity to forget of a learner in a supervised classification setting.

---

**Require:** Learner with state $Z_{t-1}$, update function $u$, inputs $X_{t:\infty}$, # of particles $N$, horizon $k$, history $H_{0:t-1}$
**Ensure:** Propensity to Forget $\Gamma_k(t)$
    Compute initial predictive distribution $q(Y_{t+k:\infty} \mid Z_{t-1}, X_{t+k-1:\infty})$
    Initialise $N$ particle copies of the learner: $\{Z_{t-1}^{(n)}\}_{n=1}^N$
    **for** each particle $n = 1$ to $N$ **do**
        **for** $s = 1$ to $k$ **do**
            Sample target $Y_{t+s}^{(n)} \sim q_f(\cdot \mid Z_{t+s-1}^{(n)}, X_{t+s-1}^{(n)})$
            Sample input $X_{t+s}^{(n)} \sim q_e(\cdot \mid H_{0:t+s-1}, Y_{t+s}^{(n)})$
            Update particle state: $Z_{t+s}^{(n)} \sim u(\cdot \mid Z_{t+s-1}^{(n)}, X_{t+s}^{(n)}, Y_{t+s}^{(n)})$
        **end for**
        Compute predictive distribution for future inputs $q_k^{(n)}(Y_{t+k:\infty} \mid Z_{t+k-1}^{(n)}, X_{t+k-1:\infty}^{(n)})$
    **end for**
    Form a mixture of the $N$ particle predictions $q_k^*(Y_{t+k:\infty} \mid Z_{t+k-1}, X_{t+k-1:\infty})$
    Compute $\Gamma_k(t) = D_{\mathrm{KL}}\left(q(Y_{t+k:\infty} \mid Z_{t-1}, X_{t+k-1:\infty}) \,\|\, q_k^*(Y_{t+k:\infty} \mid Z_{t+k-1}, X_{t+k-1:\infty})\right)$
    **return** $\Gamma_k(t)$

---

**Computational considerations.** The primary computational cost of this procedure stems from propagating $N$ independent particles over $k$ synthetic update steps, each involving a forward pass and parameter update of the learner. This can be significant for large neural networks or long consistency horizons $k$. Two sources of approximation error arise in practice. First, the predictive distributions $q_f$ and $q_k^*$ are evaluated only on a finite validation set, introducing sampling error that decreases with the number of held-out inputs. Second, the mixture $q_k^*$ is only a Monte Carlo approximation to the expectation over the learner's self-generated future. Both sources can be reduced by increasing the size of the validation set or the number of particles, though at the cost of increased computational load.

# E    ADDITIONAL RESULTS

In this section, we discuss additional results, including Bayesian learners, permutation sensitivity, and an expanded discussion of §5.

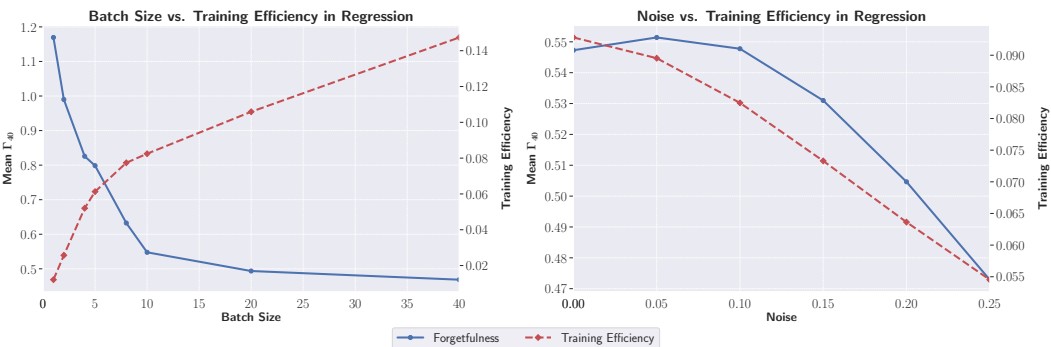

Figure 7: **Optimal forgetting is not necessarily zero.** We examine how a regression learner's training efficiency and propensity to forget vary under different hyperparameter settings. *Left:* Increasing the batch size reduces forgetting while improving training efficiency. Forgetting plateaus once the batch size approaches the dataset size (40 datapoints), at which point training efficiency increases with minimal further reduction in forgetting. *Right:* Reducing the noise in the dataset leads to substantially higher training efficiency; however, in low-noise regimes, the learner becomes more sensitive to self-consistency perturbations and therefore forgets more despite performing better on the task.

## E.1    FORGETTING IN SUPERVISED LEARNING

Here we expand on the empirical results presented in §5, examining how the dynamics of forgetting vary with model architecture, hyperparameters, and optimisation settings. In addition to the training efficiency analysis in Figure 4, we observe that hyperparameters can simultaneously shape both training efficiency and forgetting in Figure 7.

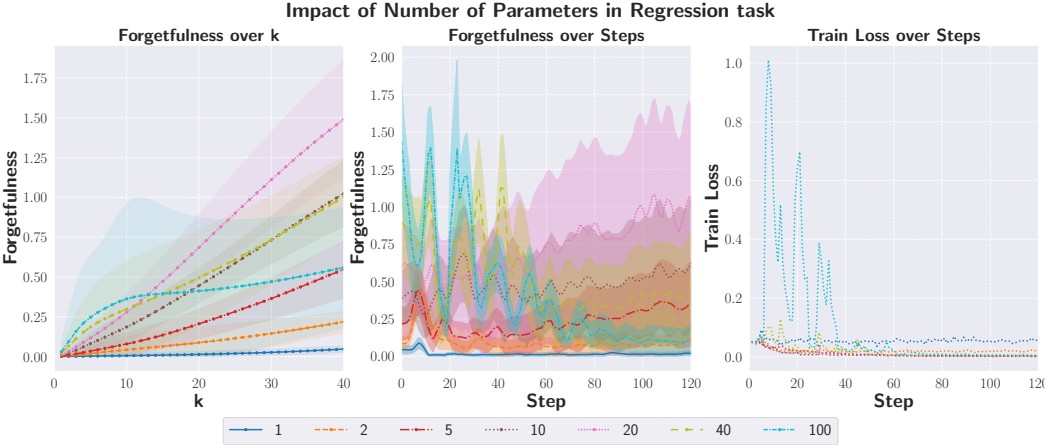

Figure 8: **Impact of model size on forgetting dynamics.** Plots illustrating the impact of varying numbers of hidden-layer parameters in a single-hidden-layer neural network on a regression task with 40 training datapoints (details in §F). *Left:* Forgetfulness as a function of the number of updates $k$, showing how the learner's propensity to forget evolves over update steps. *Middle:* Forgetfulness throughout training, highlighting changes in the learner's propensity to forget over time. *Right:* Training loss curves, comparing learning dynamics across model sizes. We observe that forgetting dynamics are strongly influenced by model size: forgetting increases with model size until the learner's parameter count approaches or exceeds the task's effective size, at which point forgetting reduces again. These dynamics have a major impact on training efficiency, as shown in Figure 4.

**Effect of model size.** Figure 8 examines how model size affects forgetting and learning efficiency on the regression task detailed in §F. Each plot compares networks with different hidden-layer sizes, trained on 40 datapoints that span both under- and over-parameterised tasks. We find that forgetting generally increases with model size, peaking near the point where the number of parameters matches the task's effective complexity, and then decreases again in the highly overparameterised task. This inverted trend mirrors the behaviour of training efficiency discussed in Figure 4.

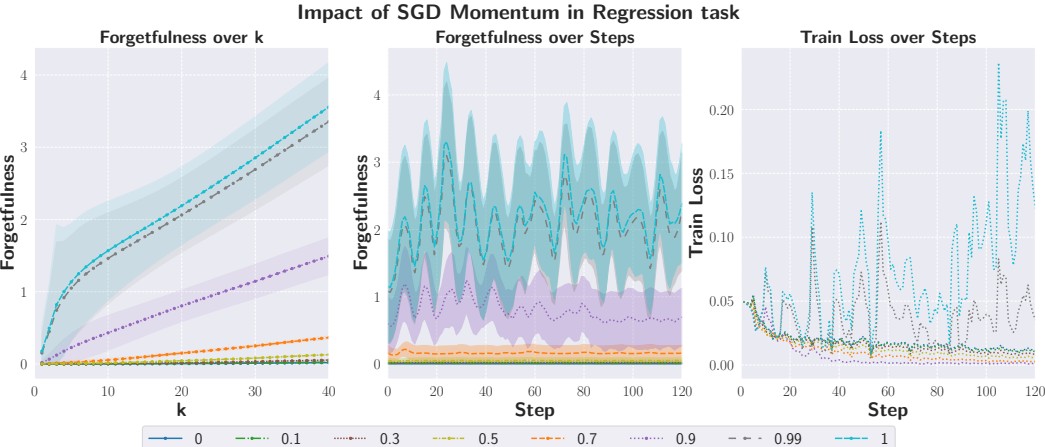

Figure 9: **Effect of the momentum parameter on forgetting dynamics.** Plots illustrating the impact of varying momentum coefficients during training of a single-hidden-layer neural network on the sinusoid regression task. *Left:* Forgetfulness as a function of the number of updates $k$. *Middle:* Forgetfulness over the course of training. *Right:* Training loss curves for each momentum value. We observe that forgetting dynamics exhibit periodicity across updates, with the mean forgetting and the oscillation amplitude increasing as momentum increases. Optimal training efficiency occurs at a momentum of 0.9, corresponding to an optimal balance of adaptation and stability.

**Effect of momentum.** In Figure 9, we vary the momentum coefficient in stochastic gradient descent while training on a regression task. The propensity to forget exhibits periodic fluctuations, increasing with higher momentum values. Both mean forgetting and the oscillatory behaviour grow with momentum, indicating a trade-off between stability and adaptivity. The most efficient training occurs around a momentum of 0.9, where moderate levels of forgetting coincide with stable and fast convergence.

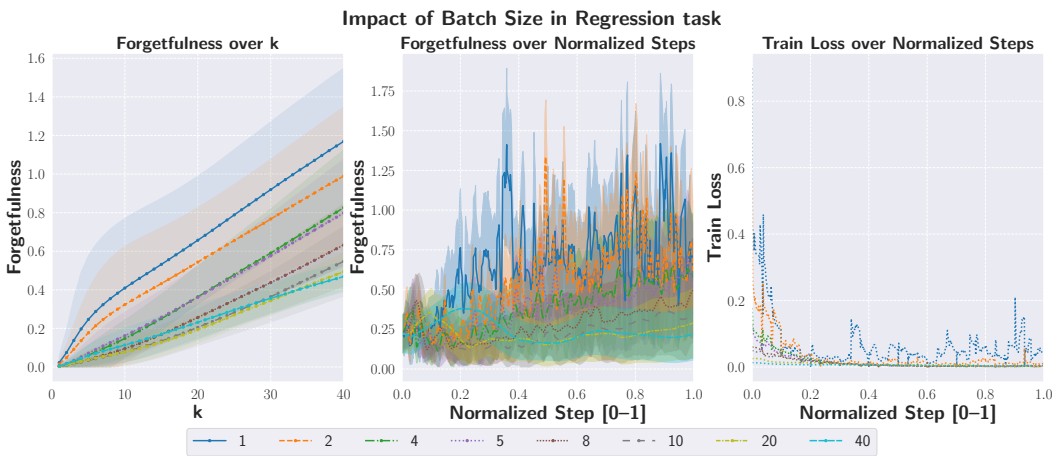

Figure 10: **Effect of batch size on forgetting dynamics.** Lines and markers indicate different batch sizes used in training a single-hidden-layer neural network on the sinusoid regression task. *Left:* Forgetfulness as a function of the number of updates $k$. *Middle:* Forgetfulness over the course of training. *Right:* Training loss curves across batch sizes. Smaller batches exhibit significant fluctuations and high variability, resulting in unstable learning and reduced efficiency. Increasing batch size stabilises learning and reduces forgetting, until the effect plateaus at a batch size of 10, beyond which forgetting remains non-zero but is approximately the same regardless of batch size.

**Effect of batch size.** Figure 10 studies the influence of batch size on forgetting dynamics. Smaller minibatches lead to higher update variability, resulting in larger oscillations and a greater propensity to forget as training progresses. As batch size increases, efficiency increases and forgetting decreases, plateauing at around one-quarter of the training dataset size. Beyond this, forgetting remains non-zero and is very similar across batch sizes greater than one-quarter of the training dataset.

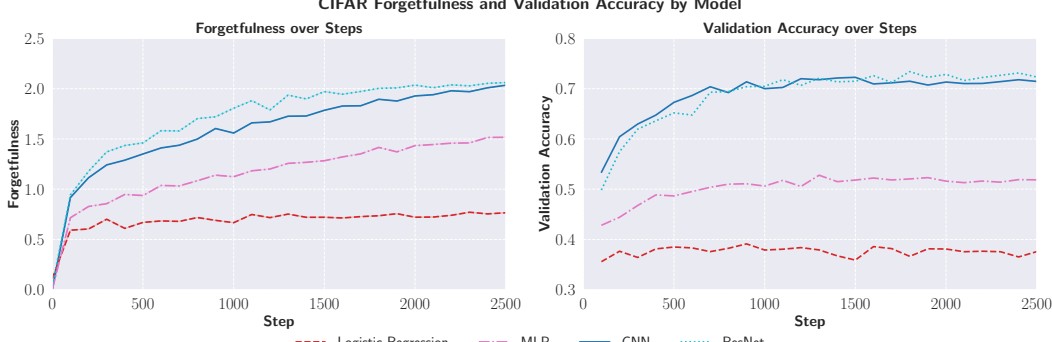

Figure 11: **Forgetting dynamics across model architectures on a high-dimensional classification task.** Different-coloured lines and marker styles denote different model types: logistic regression, MLP, CNN, and ResNet. *Left:* Forgetfulness over training updates. *Right:* Validation accuracy over training time. While CNNs and ResNets exhibit substantially higher levels of forgetting than logistic regression and MLPs (with ResNets forgetting the most), they achieve the best task performance. This implies that the most effective deep learning models do not necessarily minimise forgetting.

**Effect of architecture.** Finally, Figure 11 examines how model architecture influences forgetting on a high-dimensional classification task (CIFAR-10). We compare logistic regression, a multilayer perceptron (MLP), a convolutional neural network (CNN), and a ResNet. CNNs and ResNets exhibit considerably higher forgetting than logistic regression and MLPs, yet they achieve better task performance. This indicates that the most effective deep learning models do not necessarily minimise forgetting; instead, they maintain a balance where moderate forgetting supports continued adaptation and improved generalisation.

**Summary.** Together, these results (Figure 7, Figure 8, Figure 9, and Figure 10) show that forgetting is a pervasive property of deep learning systems, shaped by both the update dynamics and the learner's hyperparameters. Consistent with Figure 4, regimes with improved efficiency do not necessarily minimise forgetting. While high levels of forgetting can destabilise training, moderate levels appear beneficial: enabling adaptability while preserving some past information. The relationship between forgetting and learning efficiency is thus nuanced: *efficient learning requires not the absence of forgetting but its regulation.*

## E.2 FORGETTING IN REINFORCEMENT LEARNING

Here, we discuss the impact of hyperparameters in RL on forgetting dynamics.

**Buffer size.** Replay buffer size determines which past experience is retained and sampled from during Q-learning updates. A small buffer restricts the effective training distribution to the most recent transitions, while a large buffer maintains long-range temporal support.

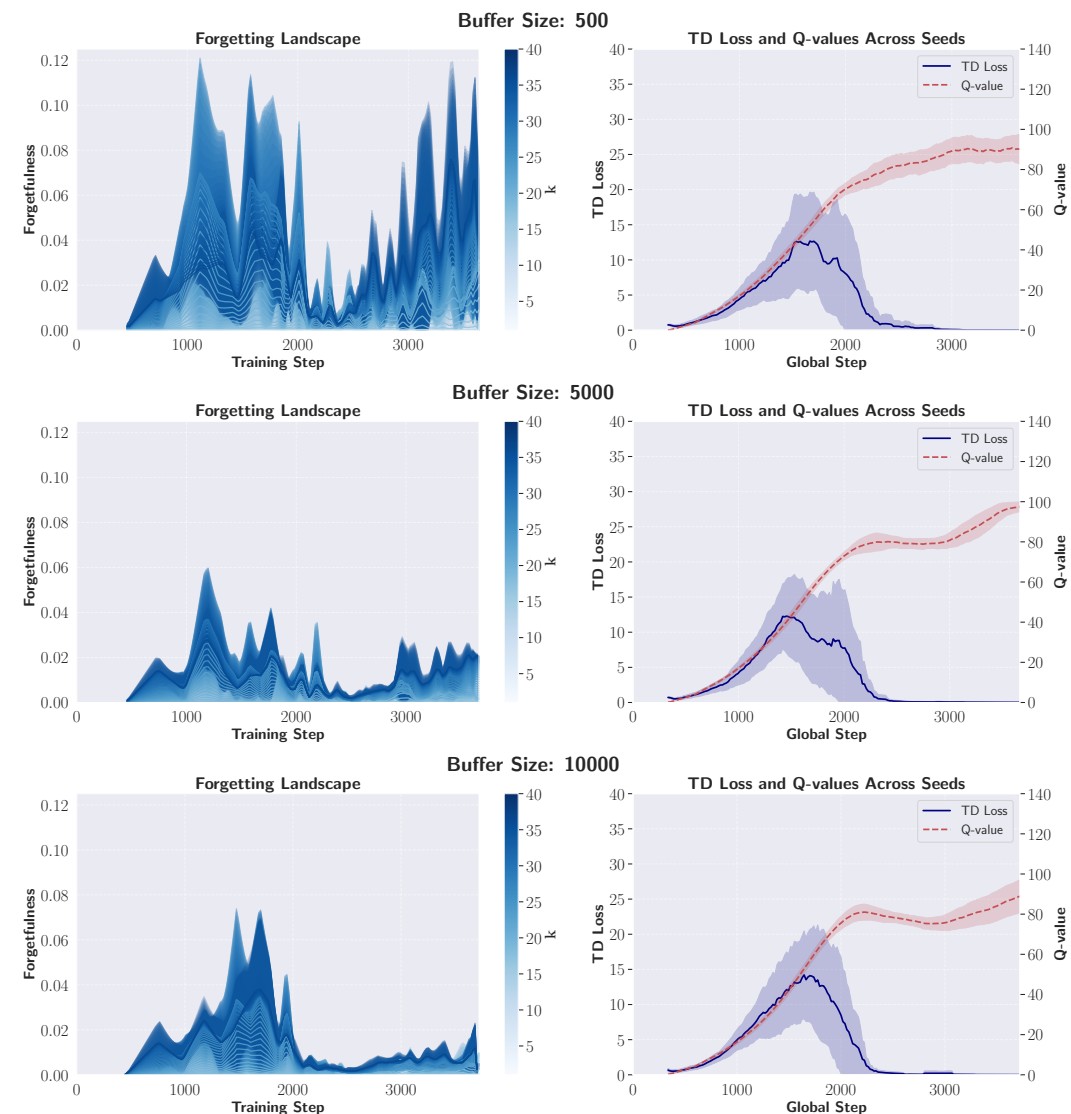

Figure 12: **Replay buffers regulate prediction support.** Forgetting landscapes for different replay buffer sizes. Small buffers produce high, unstable forgetting because the learner continually overwrites supported predictions as the data distribution shifts. Larger buffers stabilise the distribution and reduce forgetting, but overly large buffers reintroduce outdated transitions that also break self-consistency, leading to moderate forgetting and reduced learning efficiency.

With very small buffers, the training distribution shifts rapidly as old transitions are discarded. The learner, therefore, repeatedly overwrites previously supported predictions, producing large and unstable forgetting dynamics. As the buffer size increases, forgetting becomes smaller and more stable: predictions are supported by a more stable training distribution, so self-consistency violations decrease. However, extremely large buffers also reintroduce unsupported transitions, causing the learner to train on outdated targets, resulting in inconsistency. Thus, both forgetting and task performance are optimised at intermediate buffer sizes. This is observed in Figure 12.

**Target network update rate.** $\tau$ controls the target Q-network update rate. Therefore, $\tau$ controls how quickly the support for predictions evolves.

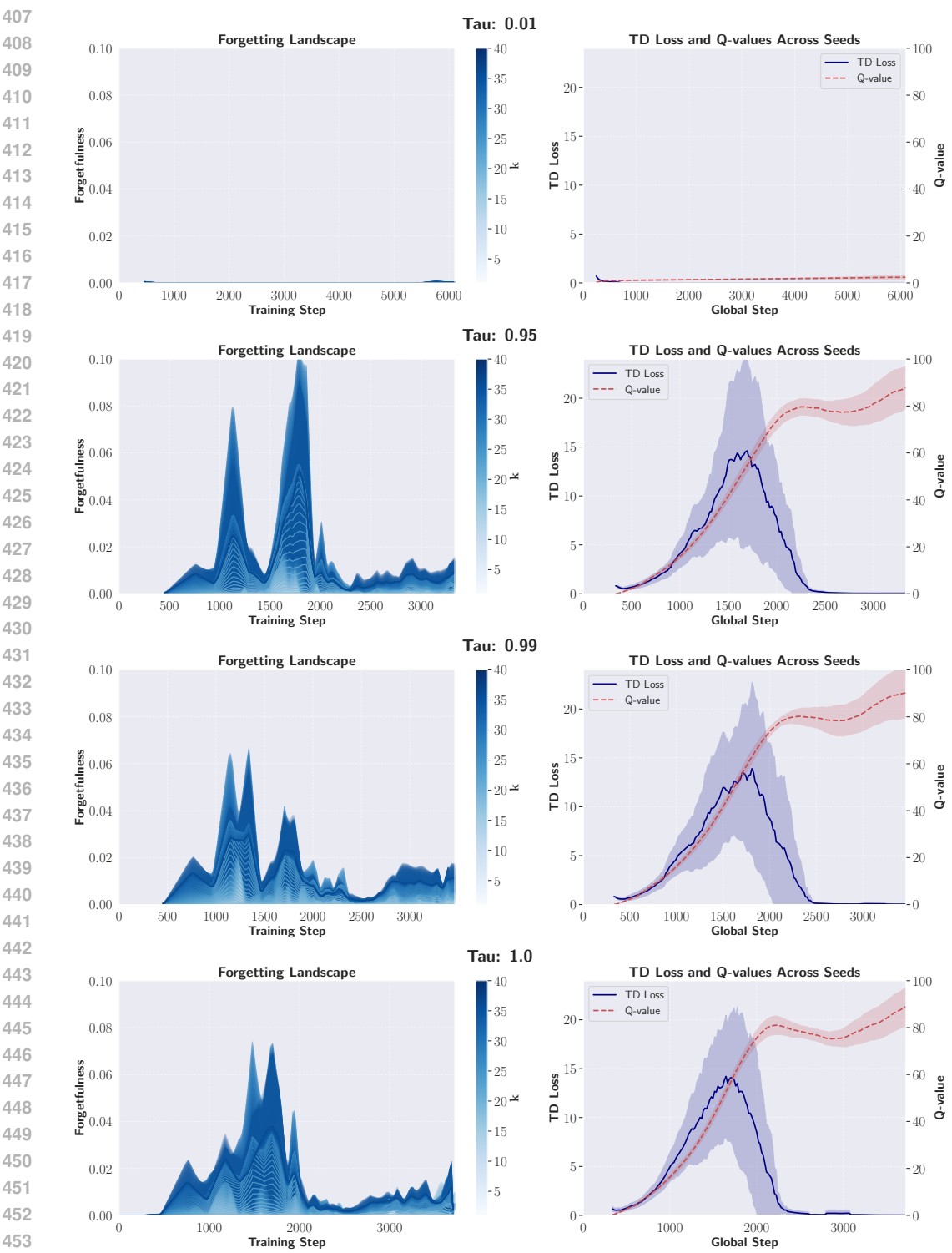

Figure 13: **Target updates trade-off stability and adaptability.** Forgetting landscapes for different target network update rates $\tau$. Large $\tau$ yields less stable forgetting dynamics due to rapidly shifting targets; small $\tau$ produces smooth but uninformative updates that prevent learning. Intermediate $\tau$ values best maintain support for predictions and thus yield balanced forgetting and effective learning.

When $\tau$ is too large, the target network is updated frequently, giving the learner a less stable reference. Predictions can become unsupported more frequently, potentially producing higher, less stable forgetting dynamics. When $\tau$ is very small, the target network is rarely updated. These result in fixed stable targets; however, they also impair the learner's ability to learn. The learner may not be forgetting, but it is also failing to learn. When $\tau$ is moderate, the target network adjusts gradually, allowing predictions at time $t$ to remain approximately supported by subsequent updates, which is important for self-consistency.

Consequently, both forgetting and performance show optimal behaviour at intermediate $\tau$: the target evolves slowly enough to provide support for predictions, yet quickly enough to adapt the fixed point toward a self-consistent solution (see Figure 13).

**Training frequency.**   Training frequency determines how often the learner applies updates relative to environment interactions. This determines the rate at which the training distribution shifts.

When the learner makes frequent updates (low training frequency values), the effective training distribution shifts very frequently. As such, updates are frequently performed on shifting distributions, leading to inconsistent predictive distributions over time.

When the learner makes moderately frequent updates, the effective training distribution is quite stable because the learner accumulates small batches of experience before updating. This creates smaller and more stable forgetting as predictions have more persistent support: the effective training distribution changes gradually, and model updates are less likely to be inconsistent.

However, when the learner makes infrequent updates, the effective training distribution again shifts frequently: the learner is updated so infrequently that each update incorporates a highly non-stationary batch of transitions. This again causes model updates to be inconsistent, again breaking support for many of the predictions the learner made earlier, and the forgetting dynamics again increase and become more unstable. In this setting, learning performance also drops because the update frequency impacts the learner's ability to learn.

Thus, optimal self-consistency emerges only at moderate training frequencies (see Figure 14).

**Summary.**   Across our RL experiments, we observe that forgetting dynamics are strongly influenced by hyperparameters such as buffer size, target network update rate, $\tau$, and training frequency. Smaller buffer sizes lead to high and unstable forgetting due to insufficient replay support; very low $\tau$ values produce stable forgetting dynamics but prevent meaningful learning; and both low and high training frequencies induce chaotic forgetting dynamics. In all cases, these dynamics previously supported by the learner's state are no longer reinforced when there is insufficient support, causing higher forgetting.

We also note that forgetting tends to track the TD loss over training, indicating that temporal difference updates implicitly regulate forgetting dynamics. These results highlight that managing forgetting is not just a consequence of network design or training heuristics, but a *vital consideration in the design of RL algorithms*: poor control of forgetting leads to unstable learning, lower sample efficiency, and suboptimal performance. Optimal algorithm design, therefore, requires balancing the reinforcement of past knowledge with the acquisition of new information.

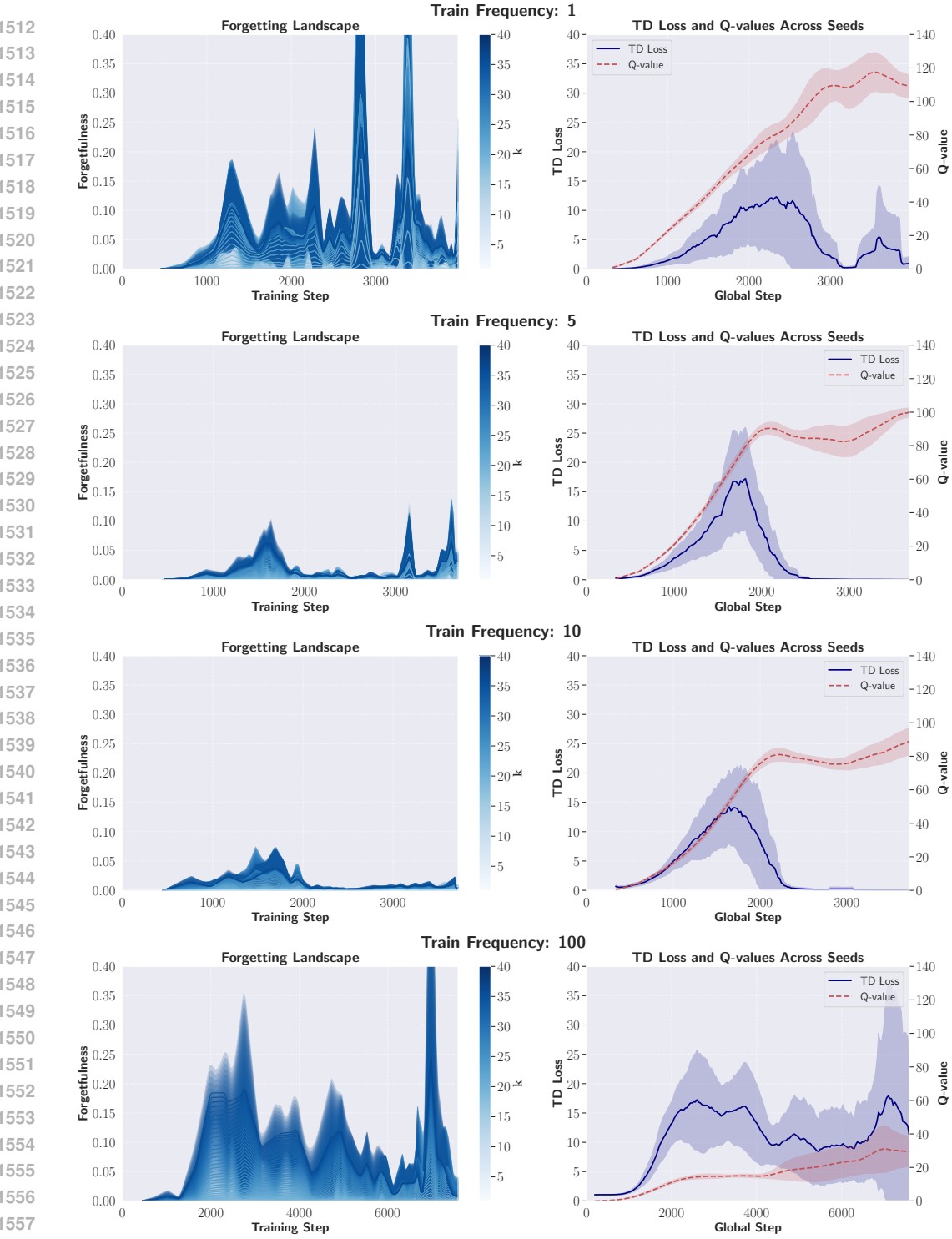

Figure 14: **Training frequency determines forgetting.** Forgetting landscapes for different training frequencies. Updating too frequently causes highly unstable forgetting due to rapid shifts in the effective training distribution. Updating too infrequently also causes large shifts in the effective training distribution. A trade-off must be found to achieve more self-consistent updates.

–appendices continue on next page–

# F  EXPERIMENT DETAILS

**Hardware.** We used NVIDIA GeForce RTX 2080 Ti GPUs to run our experiments.

**Hyperparameters.** Tables 3 and 4 lists all of the hyperparameters that were used in the experiments.

**Open-source code.** Upon acceptance, our code will be made available on GitHub.

Table 3: Hyperparameters for generative modelling and DQN reinforcement learning experiments.

| Hyperparameter | Description | Values / Type |
|---|---|---|
| **Forgetfulness settings** | | |
| k | Number of forgetfulness updates | 40 |
| num_particles | Number of particles for Monte Carlo approximation | 1000 |
| **Class Incremental-Learning** | | |
| setting | Dataset used | Two Moons |
| noise | Dataset noise | 0.1 |
| epochs | Number of training epochs | 30 |
| num_tasks | Number of tasks | 2 |
| batch_size | Batch size | 25 |
| num_samples | Number of training samples per task | 100 |
| num_val_samples | Number of validation samples | 100 |
| lr | Learning rate | 0.1 |
| optimiser | Optimiser | Adam |
| hidden_dim | Hidden layer size | 10 |
| **Reinforcement Learning (DQN)** | | |
| setting | Environment | CartPole |
| batch_size | Number of transitions sampled per gradient update | 128 |
| buffer_size | Replay buffer size | 10,000 |
| start_e | Initial exploration rate ($\epsilon$) | 1.0 |
| end_e | Final exploration rate ($\epsilon$) | 0.05 |
| exploration_fraction | Fraction of training during which $\epsilon$ decays | 0.5 |
| learning_starts | Number of steps before training starts | 10,000 |
| train_frequency | Number of steps between gradient updates | 10 |
| target_network_frequency | Frequency of hard target network updates | 500 |
| tau | Soft target update rate | 1.0 |
| gamma | Discount factor for future rewards | 0.99 |
| lr | Learning rate for optimiser | 0.00025 |
| optimiser | Optimiser used for training | Adam |
| num_eval_steps | Number of steps per evaluation | 1,000 |
| total_timesteps | Total number of environment interactions | 200,000 |
| num_parameters | Number of model parameters | 5 |

Table 4: Summary of hyperparameters and settings for all tasks. Most hyperparameters are fixed within each task, with task-specific variations highlighted where relevant.

| Hyperparameter | Description | Values / Type |
|---|---|---|
| **Regression** | | |
| setting | Dataset used | Sinusoid |
| noise | Observation noise | 0.1 |
| epochs | Number of training epochs | 30 |
| batch_size | Batch size | 10 |
| num_samples | Number of training samples | 40 |
| num_val_samples | Number of validation samples | 100 |
| lr | Learning rate | 0.1 |
| optimiser | Optimiser | Adam |
| num_parameters | Number of model parameters | 5 |
| **Classification** | | |
| setting | Dataset used | Two Moons |
| noise | Observation noise | 0.1 |
| epochs | Number of training epochs | 30 |
| batch_size | Batch size | 25 |
| num_samples | Number of training samples | 100 |
| num_val_samples | Number of validation samples | 100 |
| lr | Learning rate | 0.1 |
| optimiser | Optimiser | Adam |
| hidden_dim | Hidden layer size | 10 |
| **Generative Modelling** | | |
| setting | Dataset used | Two Moons |
| batch_size | Number of samples per training batch | 2,500 |
| epochs | Number of training epochs | 250 |
| hidden_dim | Number of hidden units in the model | 64 |
| lr | Learning rate for optimiser | 0.01 |
| optimiser | Optimiser used for training | Adam |
| noise | Noise added to dataset | 0.05 |
| num_integration_steps | Number of integration steps in the model | 100 |
| num_samples | Total number of training samples | 10,000 |
| num_val_samples | Number of validation samples | 1,000 |

