# OpenReview forum: "Forgetting is Everywhere"
_ICLR.cc/2026/Conference — Submitted to ICLR 2026_

### Official Review · Reviewer_xrmZ · 2025-10-30

**Soundness:** 3
**Presentation:** 3
**Contribution:** 2
**Rating:** 4
**Confidence:** 3

**Summary:**

This paper proposes a unified and universal theoretical framework to characterize “forgetting.” Based on the theory of “predictive distribution self-consistency,” the authors define the essence of forgetting: when learners alter their predictions upon encountering expected data, this change signifies the loss of previously acquired capabilities. Building on this concept, the paper introduces operational metrics to quantify the forgetting tendency of learning algorithms during sequential training. The authors also conduct extensive empirical experiments across multiple learning paradigms, revealing the universality of the forgetting phenomenon and the trade-off between forgetting and training efficiency.

**Strengths:**

1. The paper presents a clear and relatively general formal definition that not only explores forgetting in supervised learning but also extends to generative modeling and reinforcement learning, demonstrating the widespread nature of the forgetting phenomenon.
2. Experiments reveal that optimal training efficiency does not correspond to minimal forgetting but rather involves a trade-off, offering valuable insights for designing learning algorithms.
3. The paper is well-structured and well-written, and generally easy to follow.

**Weaknesses:**

1. Although experimental results cover multiple learning paradigms, they primarily present trends rather than delving into underlying mechanisms. For instance, in reinforcement learning, where forgetting is severe, they remain largely observational and lack analysis of reasons such as network architecture, state space, or optimizer hyperparameters.
2. While the paper presents theory and metrics, it offers limited suggestions on “how to design algorithms that reduce forgetting.” It fails to provide sufficient guidance for practitioners seeking to know “how to improve training.”

**Questions:**

1. In practical continuous learning tasks, do the metrics proposed in this paper outperform traditional indicators? How should practitioners adopt these different metrics based on prior knowledge?
2. Moderate forgetting may correspond to optimal efficiency. Is this conclusion robust across different conditions such as model architecture, dataset size, and optimization hyperparameters (e.g., momentum, learning rate)? Can more actionable recommendations be provided?
3. When learning involves significant randomness or occurs in unstable tasks, inconsistent expectations due to distribution shifts may be caused by the task itself rather than by forgetting. Does the definition of forgetting presented in this paper remain valid in such cases?

---

> ### Author Response · Authors · 2025-11-18
>
> We thank the reviewer for their thoughtful evaluation, their positive remarks on the clarity and generality of our formalism, and their recognition of the conceptual breadth of the work across supervised learning, generative modelling, and reinforcement learning. All changes made in response to the review are marked in **violet** in the revised manuscript here.
>
> Below, we address the reviewer's concerns and questions.
>
> > Experimental results primarily present trends and do not analyse underlying mechanisms.
>
> Thank you for this comment. Following the reviewer's suggestion, we have expanded the empirical discussion to provide more concrete insights. In particular:
>
> - **We broadened the analysis of reinforcement learning and supervised learning** in Figure 4, as well as providing an extensive collection of experiments in Appendix E.2. These experiments now study the dependence on batch size, data noise, and model size in supervised learning and on buffer size, training frequency, and target network update rate in reinforcement learning.
> - We improved the empirical analysis discussion (§5) to offer clearer, more interpretable takeaways and analyses rather than purely observational trends.
>
> These additions provide a substantially richer understanding of how hyperparameters affect forgetting dynamics across learning paradigms.
>
> At the same time, we emphasise that **this paper focuses on a general conceptualisation of forgetting, not on the mechanisms by which forgetting occurs**. Such mechanistic investigations depend on a multitude of factors. Our goal is to provide the foundation upon which such mechanism-level analyses can be meaningfully constructed.
>
> > The paper offers limited suggestions for reducing forgetting or improving training.
>
> We agree that developing effective CL algorithms is an important research direction. However, we believe that before we can design principled mitigation strategies, a precise conceptual understanding of forgetting is essential. Therefore, our aim is not to propose a mitigation technique or performance-oriented metric, but **to provide a general, task- and architecture-agnostic formalism that characterises forgetting in learning systems across all learning settings.**
>
> Many existing CL methods attempt to mitigate forgetting without a principled understanding of what forgetting is. Our work fills this gap. By clarifying the phenomenon at a fundamental level, **we provide a conceptual and analytical basis for developing informed algorithmic strategies in future work.**
>
> *(Response continued in next comment)*

---

> > ### Author Response · Authors · 2025-11-18
> >
> > *(Continued response)*
> >
> > **[Responses to specific questions]**
> >
> > > 1. Do the metrics in this paper outperform traditional indicators? How should practitioners adopt them?
> >
> > **We would like to clarify that our contribution is not the introduction of a performance metric. Instead, the propensity to forget is an instance of our conceptual framework that allows us to ensure our theory operates as expected.** We demonstrate that the propensity to forget behaves exactly as expected from a forgetting measure. For instance, in Fig. 3 (right), the propensity to forget increases suddenly immediately after a task change, as intuition suggests. This alignment between theory and behaviour demonstrates the reliability of the theory and the corresponding measure. Traditional metrics, on the other hand, conflate forgetting with numerous other effects, such as backward transfer. To clarify this, we have strengthened the introduction to make the point explicit.
> >
> > > 2. Is the conclusion that moderate forgetting corresponds to optimal efficiency robust under changes to architecture, dataset size, or hyperparameters?
> >
> > We thank the reviewer for this question; yes! We have **added an additional panel to the main trade-off figure** (Figure 4), and we also provide several further examples in Appendix E. Across all of the diverse conditions that we evaluate, the qualitative trade-off remains: optimal training efficiency rarely corresponds to zero forgetting in deep learning settings. This relationship is robust.
> >
> > These expanded experiments support the generality of the observation while emphasising that the exact optimum depends on the specific learning system; another reason why a general formalism for forgetting is necessary.
> >
> > > 3. Does the definition remain valid when learning is highly stochastic or when distribution shifts are induced by the task itself?
> >
> > Yes, **our definition of forgetting is explicitly designed to apply to both stationary and non-stationary settings, including those with stochasticity**. For example, in Figure 3 (right), after a task boundary, we see an abrupt increase in forgetting. The RL environments studied in §5.4 and Appendix E.2 can be considered "highly stochastic", and we see that the forgetting dynamics match those of the TD loss.
> >
> > We clarify this in the updated manuscript (lines 228-234). In non-stationary or unstable tasks, the learner may appropriately update its beliefs due to new information. Our theory cleanly distinguishes such epistemic updates from forgetting, which reflects predictive changes inconsistent with the learner's own prior knowledge. This separation is exactly why the predictive perspective is both general and powerful.
> >
> > We thank the reviewer again for their constructive and encouraging feedback. The revised manuscript now:
> >
> > - includes expanded empirical discussion and a broader set of RL and SL experiments,
> > - offers clearer and more concrete insights in the analysis,
> > - provides an additional trade-off figure and further supporting examples in the appendix,
> > - and clarifies the generality and robustness of our definition under stochasticity and distribution shift.
> >
> > We believe these revisions have improved the clarity and empirical rigour of the work.

---

> ### Author Response · Authors · 2025-11-27
>
> Dear Reviewer xrmZ,
>
> We believe our response and updated paper have addressed your concerns by
>
> - expanding our empirical discussion and experiments,
> - providing concrete insights into our results,
> - adding supporting figures and further examples to our appendix, and
> - clarifying the generality and robustness of our definition under stochasticity and distribution shift.
>
> If there's anything further we can clarify or discuss, please let us know. We welcome any additional feedback or questions you may have before the end of the discussion period.
>
> Thank you, -Authors

---

### Official Review · Reviewer_qboB · 2025-10-31

**Soundness:** 3
**Presentation:** 3
**Contribution:** 2
**Rating:** 4
**Confidence:** 4

**Summary:**

This paper proposes a general theoretical and empirical framework for understanding forgetting in machine learning systems, treating forgetting as a universal property of adaptive learners, not just in CL settings. It formalizes the forgetfulness of a system through the concept of k-step self-consistency in predictive distributions, i.e., the divergence between the learner's updated model and its hypothetical consistent model.

**Strengths:**

1. It is interesting to study forgetting issue as a general property of machine learning models. The idea is novel.

2. It is novel to measure the forgetfulness as the propensity to forget, i.e., how much the learner system's internal representation of the future drifts purely due to its own updates, independent of environmental changes.

3. The paper conducts experiments spanning multiple machine learning settings, such as regression, classification, and reinforcement learning, leading to some insightful findings.

**Weaknesses:**

1. The paper lacks the theorectical grounding to justify the notion of predictive consistency.

2. It is unclear what the direct benefits of understanding forgetfulness of a learner's sytems are, especially, how it can contribute to the downstream tasks. It would be more interesting to investigate how the understanding can help mitigate the forgetting issues.

3. The paper focuses on the analysis on a system's self consistency in predictive disributions, treating the system as a blackbox and offering limited interpretability. It doesn't look into the internal representations or parameter dynamics (e.g., drift in embedding space) contribute to forgetting, which could be more useful to explain the forgetting phenomenon.

**Questions:**

1. Can you analyze the reliablity of the forgetfulness measurement?

2. Can you compare the forgetfulness and the typical forgetting metrics (such as accuracy drop) in current CL settings?

3. What is the computational cost of evaluating the forgetfulness of a model, especially for those large scale models?

4. What are the direct benefits of understanding the forgetfulness of a model? How can it be leveraged to mitigate the forgetting issues.

---

> ### Author Response · Authors · 2025-11-18
>
> We thank the reviewer for their thoughtful and constructive feedback, as well as for the positive comments regarding the novelty of studying forgetting as a general property of learning and the breadth of empirical settings analysed. All changes made in response to the review are marked in **violet** in the revised manuscript here.
>
> Below, we address the reviewer's concerns and questions.
>
> > The paper lacks theoretical grounding to justify predictive consistency.
>
> We appreciate this comment, but respectfully disagree with the assessment. **Our definition of forgetting is grounded in predictive Bayesianism**, a well-established theoretical framework in probability and statistical learning. Following the reviewer's feedback, we have provided explicit exposition for the predictive perspective to clarify this grounding.
>
> The revised manuscript now explicitly states (lines 228-234) that:
>
> - the learner's state is characterised through its predictive distribution over futures,
> - this perspective is theoretically rigorous and does not require Bayesian learners,
> - and it provides a unified notion of knowledge that is directly applicable to neural networks, where internal parameters are not interpretable.
>
> Importantly, we also clarify that the main contribution of the paper is not to propose a new metric, nor to study the specific mechanisms that may cause forgetting in a given learning setting (e.g., distribution shift, drift in embedding space, etc.). **We believe that before any such instance of forgetting can be studied, a conceptualisation of forgetting as a fundamental learning phenomenon is necessary.** The propensity to forget is an instantiation of this conceptual framework, not the central contribution itself. Our goal is to demonstrate how forgetting can be coherently defined across all learning systems, regardless of the learning algorithm or environment. The predictive perspective, therefore, serves as the theoretical foundation enabling this general definition.
>
> This predictive viewpoint offers additional advantages: predictive statements can be validated against realised outcomes, and it is agnostic to the inner workings of the learning algorithm. Tracking the evolution of predictive distributions naturally leads to a principled definition of forgetting.
>
> We believe the improved exposition fully addresses the reviewer's concern.
>
> > It is unclear what the direct benefits are; how does understanding forgetting help downstream tasks?
>
> Thank you for raising this point. Our objective is to first understand forgetting itself. As we emphasise in the revised manuscript (lines 39-43, 103-116), we believe that before effective mitigation strategies can be developed in continual learning, a precise conceptual understanding of the phenomena must first be formulated.
>
> However, we do also provide an analysis of forgetting in simple models, where we measure the trade-off between forgetting and learning efficiency with respect to various common choices (§5, Appendix E). Expanding on this analysis is a natural and fruitful direction for future work; however, it is outside the scope of the present paper.
>
> We hope this and the accompanying paragraphs in the introduction and literature review resolve this question.
>
> > The analysis treats the learner as a black box; internal representations or parameter drift are not examined.
>
> **The model- and algorithm-agnostic theoretical framework, which considers only predictive distributions and not the inner state of a learner, is, in fact, intentional and central to our contribution.** As clarified in the revised manuscript (lines 103-116, 306-313, and §5.1), parameter drift is neither necessary nor sufficient for forgetting. Therefore, analysing parameters or embeddings cannot reliably reveal forgetting and often confounds the analysis.
>
> Our predictive perspective deliberately treats the learner as a black box because:
>
> - many learning systems (especially deep neural networks) have non-interpretable parameter spaces, and the exact same model class can be parameterised in many different ways,
> - this notion of forgetting places minimal requirements on our access to the internal mechanisms of a learner: we are still able to define, measure, and reason about a learner’s propensity to forget in the absence of experimental access to a learner’s latent state;
> - forgetting must be defined in a way that applies across learning algorithms, and because there are learning algorithms that do not have parameters, a parameter-based definition is not a general definition, and
> - predictive distributions provide an interpretable and model-agnostic representation of the learner's knowledge.
>
> Thus, the black-box approach is a beneficial feature of our perspective, not a limitation, enabling a definition of forgetting that is universal.
>
> *(Response continued in next comment)*

---

> > ### Author Response · Authors · 2025-11-18
> >
> > *(Continued response)*
> >
> > **[Responses to specific questions]**
> >
> > > 1. Can you analyse the reliability of the forgetfulness measurement?
> >
> > This is a great question; yes! We demonstrate that the propensity to forget behaves exactly as expected from a measure of forgetting. For instance, in Fig. 3 (right), the propensity to forget increases suddenly immediately after a task change, matching intuition. This alignment between theory and behaviour demonstrates the reliability of the theory and the corresponding measure.
> >
> > > 2. Can you compare forgetfulness to typical metrics, such as accuracy drop?
> >
> > Metrics such as accuracy drop are useful when the data is split into tasks (for example, class-incremental learning or other instances of distribution shift). They would not allow us to measure forgetting in a stationary supervised learning setting.
> >
> > We discuss the difference between forgetting and typical metrics, such as the accuracy drop in lines 103-117. In short, these metrics do not isolate forgetting; they conflate it with backward transfer, task difficulty, and distribution shift. Our conceptualisation is therefore complementary and foundational, rather than a replacement for such metrics.
> >
> > > 3. What is the computational cost for large models?
> >
> > Thank you for raising this question! We have now **added a computational and approximation considerations section in Appendix D** (lines 1179-1186). In general, because the approach requires Monte Carlo approximation, the measure scales in computational complexity with larger models or NFE (when discussing flow matching/diffusion models). This also introduces approximation errors. This computation can be made more efficient by using more sophisticated sampling techniques.
> >
> > > 4. What are the benefits of understanding forgetfulness? How can this mitigate forgetting?
> >
> > The manuscript now emphasises that understanding forgetting is the necessary first step toward mitigating it (lines 39-43), and has discussed the implications of these misunderstandings on past work (lines 108-117). We believe that **to develop effective future algorithms that optimally utilise forgetting, a precise conceptual understanding of forgetting must be formulated first**. In short, it is important that we have precise, well-understood concepts that organise our science so that the hypotheses we form and the theories we devise are rooted in a clear foundation.
> >
> > Therefore, our aim is to provide a general formalism that characterises forgetting in learning systems. **Our work provides the conceptual and analytical basis for future methods of mitigation to be built upon.**
> >
> > We thank the reviewer again for their insightful comments. The revised manuscript now:
> >
> > - strengthens the theoretical grounding for predictive consistency,
> > - clarifies that the paper offers a conceptual framework, not a metric-focused contribution,
> > - highlights the value of a predictive, architecture-agnostic definition of forgetting, and
> > - includes computational considerations.
> >
> > We believe these revisions have helped improve both the clarity and the impact of the work while preserving its conceptual contributions.

---

> ### Author Response · Authors · 2025-11-27
>
> Dear Reviewer qboB,
>
> We believe our response and updated paper have addressed your concerns by
>
> - responding to your specific questions,
> - clarifying the theoretical foundation of the predictive consistency,
> - highlighting the value of a predictive, architecture-agnostic definition of forgetting, and
> - discussing the computational considerations of the propensity to forget measure.
>
> If there's anything further we can clarify or discuss, please let us know. We welcome any additional feedback or questions you may have before the end of the discussion period.
>
> Thank you, -Authors

---

### Official Review · Reviewer_LfRt · 2025-10-31

**Soundness:** 3
**Presentation:** 3
**Contribution:** 2
**Rating:** 4
**Confidence:** 3

**Summary:**

This paper proposes a unified theoretical framework to explain forgetting across diverse learning settings, including Reinforcement Learning, Continual Learning, and Generative Modeling. The authors argue that forgetting should be viewed separately from prediction accuracy and instead defined as a change in the induced future. A metric named “propensity to forget” is introduced to measure this change, along with empirical studies to support the theory.

**Strengths:**

- The paper is generally well written and organized (though a few statements may need improvement).

- The work introduces a novel conceptual angle by defining forgetting based on induced futures rather than accuracy degradation.

**Weaknesses:**

- The statement in Lines 45–47 appears too strong (and several other statements in the paper with this assumption). The claim assumes that the learner does not gain any new information. However:

  - How can we ensure the learner indeed learns nothing new during such updates, especially under stochasticity?
  - If the data distribution changes (e.g., under distribution shift or class-incremental settings), changes in induced futures may come from newly acquired knowledge, not forgetting.

  This suggests that the theory may be limited to stationary environments or require clear assumptions about what constitutes “new information.”

- The analysis seems largely limited to forgetting caused by the stochastic nature of training (e.g., momentum changes in Figure 2), rather than more widely studied sources of forgetting, such as task interference.

- Some concepts are vague and would benefit from clearer definitions:

  - Lines 216–219: What exactly is meant by “performance”? Because there are metrics that incorporate forgetting and accuracy (e.g, Backward Transfer)

  - Line 242: Clarify what is meant by “exposed”: does this refer to training or inference?

  - Line 242: What does “data it already expects” refer to: historical data, or future trajectories?

- It would strengthen the paper to include a concrete algorithm or procedural guide to compute Equation (9).

**Minor:**

- Although Y is defined, it is sometimes confusing with ground-truth labels (e.g., Line 128).
- Line 130: Next observations do not always depend on history (e.g., i.i.d).

**Questions:**

Please refer to the weakness section.

---

> ### Author Response · Authors · 2025-11-18
>
> We thank the reviewer for their feedback and positive remarks on the paper's organisation and our predictive perspective on forgetting. All changes in response to the review are marked in violet in the revised manuscript.
>
> Below, we address the reviewer's concerns.
>
> > How can we ensure no new information is learned? What if the data distribution shifts? If the data distributions change, differences in induced futures may reflect new knowledge rather than forgetting.
>
> Thank you for raising these important conceptual points. Based on this and similar comments from other reviewers, we realised that our predictive Bayesian formalisation required clearer exposition. We have now clarified and provided background for this material.
>
> We **added a dedicated paragraph** (lines 228-234) explaining that:
>
> - our notion of induced futures is based on predictive Bayes, a standard tool for analysing the knowledge state of any learner, not just Bayesian algorithms,
> - and predictive Bayes does not assume stationarity; it applies equally to stationary, non-stationary,  task-structured, and unstructured data.
>
> Crucially, because induced futures are generated from *the learner's own predictive distribution* (§3.2), simulated updates do not introduce new environment information. This ensures that any subsequent change in predictions *cannot be attributed to newly acquired data*. Thus, forgetting is formally defined as a change in predictive beliefs conditioned on the same information the learner already possesses (§4.2), unlike changes that occur during ordinary training.
>
> **We hope this clarifies that the formalism we introduce naturally accommodates stochasticity, distribution shift, and non-stationary settings.** We empirically evaluate each of these settings and show that our conceptualisation of forgetting exhibits the intuitive behaviour we would expect (§5 and Appendix E).
>
> To prevent misinterpretation, we **added further clarifying text on the scope and limits of the theory** (lines 352-359). We believe these additions fully address the reviewer's concerns and resolve the earlier ambiguity.
>
> > Inclusion of an algorithm; concept clarity.
>
> We appreciate the suggestion to include a concrete algorithm for computing the propensity to forget. We have now added the algorithm to the appendix (lines 1160-1177), along with an accompanying discussion to guide readers.
>
> We also revised several passages for clarity in response to the reviewer's comments:
>
> 1. The issue raised regarding lines 216-219 is now addressed at lines 270-274,
> 2. The ambiguities in Line 242 are resolved at lines 296-300.
>
> We thank the reviewer again for their helpful feedback. The revised manuscript now:
>
> - provides a clearer and more detailed explanation of the predictive Bayesian foundation underlying predictive distributions / induced futures,
> - explicitly shows that our definition of forgetting naturally accommodates distribution shift and non-stationary settings,
> - and includes a concrete algorithm for computing the propensity to forget.
>
> We believe these changes improve the clarity and accessibility of the work while preserving its conceptual contributions.

---

> > ### Author Response · Authors · 2025-11-27
> >
> > Dear Reviewer LfRt,
> >
> > We believe our response and updated paper have addressed your concerns by
> >
> > - providing a detailed discussion of the predictive Bayesian foundation underlying predictive distributions / induced futures,
> > - explicitly showing that our definition of forgetting naturally accommodates distribution shift and non-stationary settings,
> > - including a concrete algorithm for computing the propensity to forget, and
> > - revising several passages for clarity in response to your comments.
> >
> > If there's anything further we can clarify or discuss, please let us know. We welcome any additional feedback or questions you may have before the end of the discussion period.
> >
> > Thank you, -Authors

---

### Official Review · Reviewer_GaRd · 2025-11-01

**Soundness:** 3
**Presentation:** 2
**Contribution:** 2
**Rating:** 4
**Confidence:** 4

**Summary:**

This paper proposes a unified theoretical framework for forgetting, defining forgetting as the inconsistency of a mode's predictive distribution under self-consistent conditions. Starting from the interaction between a learner and its environment, the authors establish a general probabilistic process model and derive a computable measure of forgetting. Through experiments on regression, classification, generative modeling, continual learning, and reinforcement learning, they show that forgetting is ubiquitous across paradigms and that a moderate amount of forgetting can actually enhance learning efficiency.

**Strengths:**

Strengths

1. The paper is theoretically innovative, proposing a task-agnostic and algorithm-agnostic definition of forgetting that could serve as a foundational framework for theoretical analysis in the community.

2. The mathematical derivations are rigorous and the concepts are clearly defined.

3. The experiments cover multiple learning paradigms.

4. The finding that "moderate forgetting leads to optimal learning efficiency" is insightful.

**Weaknesses:**

Weaknesses
1. First, the "powerful insight" stated in the introduction is, while correct, rather self-evident and perhaps too simple to be considered an insight. In deep learning, it is well known that parameter updates via back-propagation naturally change the model's internal representations and thereby its predictions on data, which in turn causes forgetting. This seems more like a direct consequence of parameter modification than a novel conceptual observation.

2. The paper presents basic yet comprehensive theoretical analyses, but it remains unclear how much new understanding or inspiration these analyses bring to the community. Valuable conclusions only appear starting from Section 4, and this part is relatively short. I noticed that the appendix contains additional experiments and some impressive findings; some of them could be moved into the main text, possibly reducing earlier theoretical exposition. Even then, the empirical conclusions still seem limited in value — for example, Takeaway 4.1 is an interesting idea, but the supporting experiments are overly simple; merely changing the momentum in SGD is insufficient to substantiate the claim. Moreover, the discussion in Takeaway 4.2 is somewhat vague: I do not fully understand its connection to real-world reinforcement learning settings or what concrete insights it offers.

**Questions:**

See weakness

---

> ### Author Response · Authors · 2025-11-18
>
> We thank the reviewer for their thoughtful feedback and recognition of the paper's theoretical novelty, rigour, breadth, and ambition. All changes in response to the reviewer's comments are marked in violet in the revised manuscript.
>
> Below, we address the reviewer's concerns in detail.
>
> > The insight seems self-evident; parameter updates naturally cause forgetting.
>
> We appreciate this observation. We agree that, in deep learning, parameter updates often induce forgetting. However, one of our contributions is to **define forgetting as a phenomenon distinct from its mechanisms**: parameter changes may cause forgetting, but forgetting is not equivalent to those changes. Not all learning algorithms that can forget have parameters; some parameter changes are due to learning or are benign, and Bayesian models can update their parameters without forgetting; therefore, parameter updates are neither necessary nor sufficient for forgetting.
>
> To make this distinction explicit, we have added the following:
>
> - **New introduction paragraph (lines 39-43):** Emphasises that forgetting is a property of belief change in any learning system, not a property of parameter updates, policy updates, or performance loss.
> - **New paragraph in related work (lines 103-116):** Highlights how equating forgetting with parameter or policy drift has obscured conceptual clarity in the literature,
> - **New subsection in empirical analysis (§5.1):** Presents a learner that updates parameters yet never forgets, which demonstrates that parameter changes alone do not imply forgetting.
>
> With these modifications, we hope to have better motivated our work and distinguish it from the existing literature.
>
> > Empirical conclusions seem limited; takeaways do not provide concrete insights.
>
> Thank you for this feedback. We have substantially expanded the empirical analysis:
>
> - **Empirical analysis section expanded**: Introduced a subsection on unforgetful learners and further clarified the implications of each result.
> - **New elbow plots (lines 468-485):** Improved the forgetting-learning efficiency trade-off plots so that the results are easier to interpret. This illustrates that the optimal level of forgetting is not always zero in deep learning.
> - **Additional hyperparameter examples:** Demonstrate that the trade-off generalises beyond momentum to batch size, noise in the data, and model size (lines 468-485 in the main text; lines 1192-1210 in the appendix).
> - **Clarified takeaways:** Each takeaway (boxed throughout the text) now highlights concrete insights on forgetting, its dynamics, common misconceptions, and its relationship to learning efficiency.
> - **Additional RL experiments:** Added to Appendix E.2, with accompanying analysis to clarify the connection to practical RL settings.
>
> We thank the reviewer again for the constructive feedback. We believe that the suggested improvements implemented have significantly enhanced the paper. The revised manuscript now:
> - clarifies the conceptual novelty of separating forgetting from parameter updates,
> - strengthens the empirical analysis,
> - provides a clearer connection to RL and general learning settings.
>
> These revisions highlight the paper's novelty and impact while maintaining its strong theoretical foundations.

---

> ### Author Response · Authors · 2025-11-27
>
> Dear Reviewer GaRd,
>
> We believe our response and updated paper have addressed your concerns by
>
> - clarifying the conceptual implications of our insight,
> - distinguishing between forgetting and parameter updates,
> - strengthening the empirical analysis, and
> - providing a more apparent connection to RL and general learning settings.
>
> If there's anything further we can clarify or discuss, please let us know.
> We welcome any additional feedback or questions you may have before the end of the discussion period.
>
> Thank you, -Authors

---

### Author Response · Authors · 2025-11-18
**Response to All Reviewers**

We thank the reviewers for their feedback, which improved the paper’s presentation, empirical section, and clarity of scope. All changes are marked in violet in the revised manuscript.

Overall, we were encouraged by the reviewers' enthusiasm regarding the ambition, clarity, and novelty of the work, as well as the recognition that a principled, task-agnostic conceptualization of forgetting "could serve as a foundational framework for theoretical analysis in the community." The reviews raised excellent questions that helped us identify where the scope and intent of the paper were insufficiently clear, and motivated us to strengthen the empirical analysis and refine key assumptions.

Across the reviews, we observed four recurring themes:

1. Requests for increased empirical depth and stronger takeaways.
2. Confusion regarding the role of parameter/belief updates in forgetting.
3. Uncertainty about whether the formalism applies beyond stationary settings.
4. Comments about forgetting in CL vs. forgetting more generally.

We address each of these concerns below.

1. **Expanded empirical section.**

To address requests for more empirical depth, we:

- added a subsection on unforgetful learners demonstrating that parameter changes do not imply forgetting (§5.1),
- improved the momentum-forgetting figure to show the forgetting-training efficiency trade-off plots, and added another plot to show the generality of this proposition figure (lines 468-485, 1193-1210),
- strengthened the takeaways (lines 350, 422, 465, 520) such that they offer more concrete insights,
- added additional reinforcement learning (RL) plots in the appendix to demonstrate the importance of forgetting in RL when changing hyperparameters (Appendix E.2),
- included an algorithm for computing the propensity to forget in the appendix (lines 1160-1177).

2. **Parameter changes do not imply forgetting.**

We clarify that **forgetting and belief/parameter updates are distinct phenomena**: Not all changes to parameters entail forgetting. We have added a discussion of this point in the related work (lines 103-116, 306-313) and empirically demonstrated it in a new section: the unforgetful Bayesian learner (§5.1). This misconception further **demonstrates the need for a comprehensive definition and conceptualisation of forgetting** that applies to all learning settings and learners, regardless of how they are implemented.

3. **Theoretical foundation and stationarity.**

The reviewers made us realise that we had insufficiently explained the predictive Bayesian perspective we took in the paper. We therefore added a dedicated subsection (lines 228-234) clarifying that predictive Bayes underlies our notion of the induced futures/predictive distribution and that it:

- is a standard and well-established theoretical tool,
- does not assume Bayesian learners (as predictive Bayes is a perspective on knowledge, not a restriction on learning algorithm), and
- does not require stationarity.

To further clarify the scope, we added text to the paper  (lines 352-359) that explicitly details the limits of the theory. Overall, our theory applies equally to stationary, non-stationary, task-structured, and unstructured settings.

4. **Forgetting as a general property of learning**

Our goal is to **study forgetting as a fundamental property of learning**, independent of any particular domain or architecture. The learning setting we adopt (CL, RL, or otherwise) and the specific class of learner (neural networks, Bayesian, and so on) serve as instances of a broader phenomenon. Our aim was to formulate a general theory of forgetting that can accommodate all of these settings. In light of the reviewers’ feedback, we have **added a new paragraph to the introduction (lines 39-43) as well as to the related work section (lines 103-116)**, which motivates the need for a general, task- and algorithm-agnostic conceptualisation of forgetting.

We view this generality as a strength of the work, as it enables broad analysis and framing across learning settings and algorithmic approaches.

**We believe these clarifications and additions have substantially strengthened the paper. Below, we address each reviewer's comments in detail.**

---

### Meta-Review · Area_Chair_8keF · 2026-01-05

**Summary:**

This paper received four reviews, all with an initial rating of 4 (= marginally below the acceptance threshold). The authors provided a rebuttal to each of the four reviews, but unfortunately none of the four reviewers responded to this rebuttal before the premature end of the discussion phase.

If there had been a full discussion phase, I expect that most of the reviewers would not have raised their rating.
The main reason for this is that I think that the main concern raised by (most of the) reviewers has not been convincingly addressed in the rebuttal.

I think the reviewer’s main concern is nicely phrased by reviewer GaRd: “The paper presents basic yet comprehensive theoretical analyses, but it remains unclear how much new understanding or inspiration these analyses bring to the community.” A similar sentiment is expressed by Reviewer qboB: “It is unclear what the direct benefits of understanding forgetfulness of a learner's sytems are, especially, how it can contribute to the downstream tasks.” My interpretation of the reviews is that the reviewers appreciated the rigorous approach of the paper and the original perspective it takes, but that for none of the reviewers it is clear what concrete contribution this paper is making or how the insights from this paper could inform further research.

My own interpretation after reading the paper is similar. I generally enjoyed reading the paper, in particular I enjoyed some of its conceptual reasoning and arguments, but I was not able to pinpoint important concrete new insights or take-aways.

In my reading, the authors’ rebuttal does not succeed in providing or clarifying such important concrete new insights or take-aways.

**Reviewer Concerns:**

As discussed in more detail above, I think the main outstanding concern is that the paper does not clearly articulate the concrete contribution this paper is making or how the insights from this paper could inform further research.

**Reviewer Scores:**

As discussed in more detail above, I expect that most of the reviewers would not have raised their score if they had been able to participate fully in the discussion.

---

### Decision · Program_Chairs · 2026-01-26

Reject